# Neuron-associated macrophage proliferation in the sensory ganglia is associated with peripheral nerve injury-induced neuropathic pain involving CX3CR1 signaling

**Rafaela M Guimarães**[1,2†], **Conceição E Aníbal-Silva**[1,2†], **Marcela Davoli-Ferreira**[1,2,3], **Francisco Isaac F Gomes**[1], **Atlante Mendes**[1], **Maria CM Cavallini**[1,2], **Miriam M Fonseca**[1,4], **Samara Damasceno**[1], **Larissa P Andrade**[1,2], **Marco Colonna**[5], **Cyril Rivat**[6], **Fernando Q Cunha**[1], **José C Alves-Filho**[1], **Thiago M Cunha**[1]*

[1]Center for Research in Inflammatory Diseases (CRID), Department of Pharmacology, Ribeirão Preto Medical School, University of São Paulo, Ribeirao Preto, Brazil; [2]Graduate Program in Basic and Applied Immunology, Ribeirão Preto Medical School, University of São Paulo, Ribeirão Preto, Brazil; [3]Department of Physiology and Pharmacology, Snyder Institute for Chronic Diseases, Cumming School of Medicine, University of Calgary, Calgary, Canada; [4]Department of Anesthesiology, Pain Mechanisms Laboratory, Wake Forest University School of Medicine, Winston-Salem, United States; [5]Department of Pathology and Immunology, Washington University School of Medicine in Saint Louis, Saint Louis, United States; [6]Univ Montpellier, Montpellier, France; Inserm U-1298, Institut des Neurosciences de Montpellier, Montpellier, France

*For correspondence: thicunha@fmrp.usp.br

†These authors contributed equally to this work

Competing interest: The authors declare that no competing interests exist.

**Abstract** Resident macrophages are distributed across all tissues and are highly heterogeneous due to adaptation to different tissue-specific environments. The resident macrophages of the sensory ganglia (sensory neuron-associated macrophages, sNAMs) are in close contact with the cell body of primary sensory neurons and might play physiological and pathophysiological roles. After peripheral nerve injury, there is an increase in the population of macrophages in the sensory ganglia, which have been implicated in different conditions, including neuropathic pain development. However, it is still under debate whether macrophage accumulation in the sensory ganglia after peripheral nerve injury is due to the local proliferation of resident macrophages or a result of blood monocyte infiltration. Here, we confirmed that the number of macrophages increased in the sensory ganglia after the spared nerve injury (SNI) model in mice. Using different approaches, we found that the increase in the number of macrophages in the sensory ganglia after SNI is a consequence of the proliferation of resident CX3CR1+ macrophages, which participate in the development of neuropathic pain, but not due to infiltration of peripheral blood monocytes. These proliferating macrophages are the source of pro-inflammatory cytokines such as TNF and IL-1b. In addition, we found that CX3CR1 signaling is involved in the sNAMs proliferation and neuropathic pain development after peripheral nerve injury. In summary, these results indicated that peripheral nerve injury leads to sNAMs proliferation in the sensory ganglia in a CX3CR1-dependent manner accounting for neuropathic pain development. In conclusion, sNAMs proliferation could be modulated to change pathophysiological conditions such as chronic neuropathic pain.

## Editor's evaluation

Guimaraes et al. address the origin of the macrophage increase in sensory ganglia after peripheral nerve injury. The authors show that there is no major influx by blood-derived monocytes into ganglia after injury and that resident macrophages proliferate, which is dependent on CX3CR1 signaling. Overall the work is clear and sound and should be of interest to immunologists and neurobiologists.

## Introduction

Most organs across the body contain tissue-resident populations of macrophages (*Wynn et al., 2013*; *Perdiguero and Geissmann, 2016*). Historically, these resident cells are well-known for participating in host defense and in the clearance of dead cells and tissue debris, contributing to a range of pathological processes (*Ginhoux and Guilliams, 2016*; *Davies et al., 2013*; *Wynn and Vannella, 2016*). It has been appreciated that beyond their classical role in inflamed tissues, macrophages are a heterogeneous population, exhibiting high functional plasticity that is correlated with the specific functions of each tissue and niche in which they reside (*Epelman et al., 2014*; *Van Hove et al., 2019*).

Among the distinct subsets of tissue-resident macrophages, those that colonize the peripheral nervous system (PNS) have only recently been studied in more detail. In the PNS, macrophages distributed along the sciatic nerve and in the sensory ganglia (e.g. dorsal root ganglia - DRG and trigeminal ganglion) are in close contact with the primary sensory neurons, known as (sNAMs *Silva et al., 2021*; *Kolter et al., 2020*). These resident macrophages play a crucial role in nervous tissue repair (*Feng et al., 2023*) and, more recently, they have been described as an important component in the development of neuropathic pain caused by peripheral nerve injury either in male and female mice (*Guimarães et al., 2019*; *Yu et al., 2020*).

In the injured peripheral nerves, the sNAMs functions are associated with phagocytosis of cell debris and release of early inflammatory mediators, which in turn contribute to the recruitment of neutrophils and CCR2-expressing monocytes (*Calvo et al., 2012*; *Kim and Moalem-Taylor, 2011a*; *Lindborg et al., 2017*; *Jung et al., 2009*). These changes propagate the neuroinflammatory response at the level of sensory ganglia characterized by activation of glial cells and sNAMs and the consequent release of pro-inflammatory cytokines, in particular, IL-6, IL-1β, and TNF (*Yu et al., 2020*; *Kwon et al., 2013*; *Ji et al., 2014*).

Several studies using different peripheral nerve injury models have described an increase in the number of macrophages/monocytes surrounding the cell body of sensory neurons in the DRGs (*Kwon et al., 2013*; *Liu et al., 2010*; *Kim et al., 2011b*; *Huang et al., 2014*; *Kallenborn-Gerhardt et al., 2014*; *Luo et al., 2019*; *Kalinski et al., 2020*). Although these studies have referred to this increase as a result of the infiltration of peripheral blood monocytes, recent studies have suggested that resident sNAMs, in the DRGs, also proliferate after peripheral nerve injury (*Iwai et al., 2021*; *Yu et al., 2020*). Thus, further studies are still necessary to elucidate the real contribution of these events to the increase of macrophages in the sensory ganglia after peripheral nerve injury. In the present study, by using a combination of different approaches, our findings indicate that after peripheral nerve injury, the rise in the macrophage number in the sensory ganglia (e.g. DRGs) is essentially due to the proliferation of local/resident CX3CR1+ macrophages rather than the recruitment of blood circulating monocytes, and that this proliferation process is associated with nerve injury-induced pain hypersensitivity. Furthermore, we showed that CX3CR1 signaling on sNAMs mediates their proliferation/activation triggering pro-inflammatory cytokines synthesis.

## Results

### SNI triggers an increase in the number of macrophages in the DRGs

Evidence shows that peripheral nerve injury induces an increase in the number of macrophages in the sensory ganglia (DRGs and trigeminal ganglia) (*Yu et al., 2020*; *Kwon et al., 2013*; *Iwai et al., 2021*). Herein, we sought to characterize this process using a classical model of peripheral nerve injury, SNI in mice (*Figure 1A*; *Decosterd and Woolf, 2000*). Firstly, after nerve injury, we evaluated the gene expression profile of macrophage markers, such as *Aif1* (Iba1) and *Csf1r* (CSF1R), in the ipsilateral DRGs at different time points (*Figure 1B*). We found that *Aif1* and *Csf1r* genes expression were

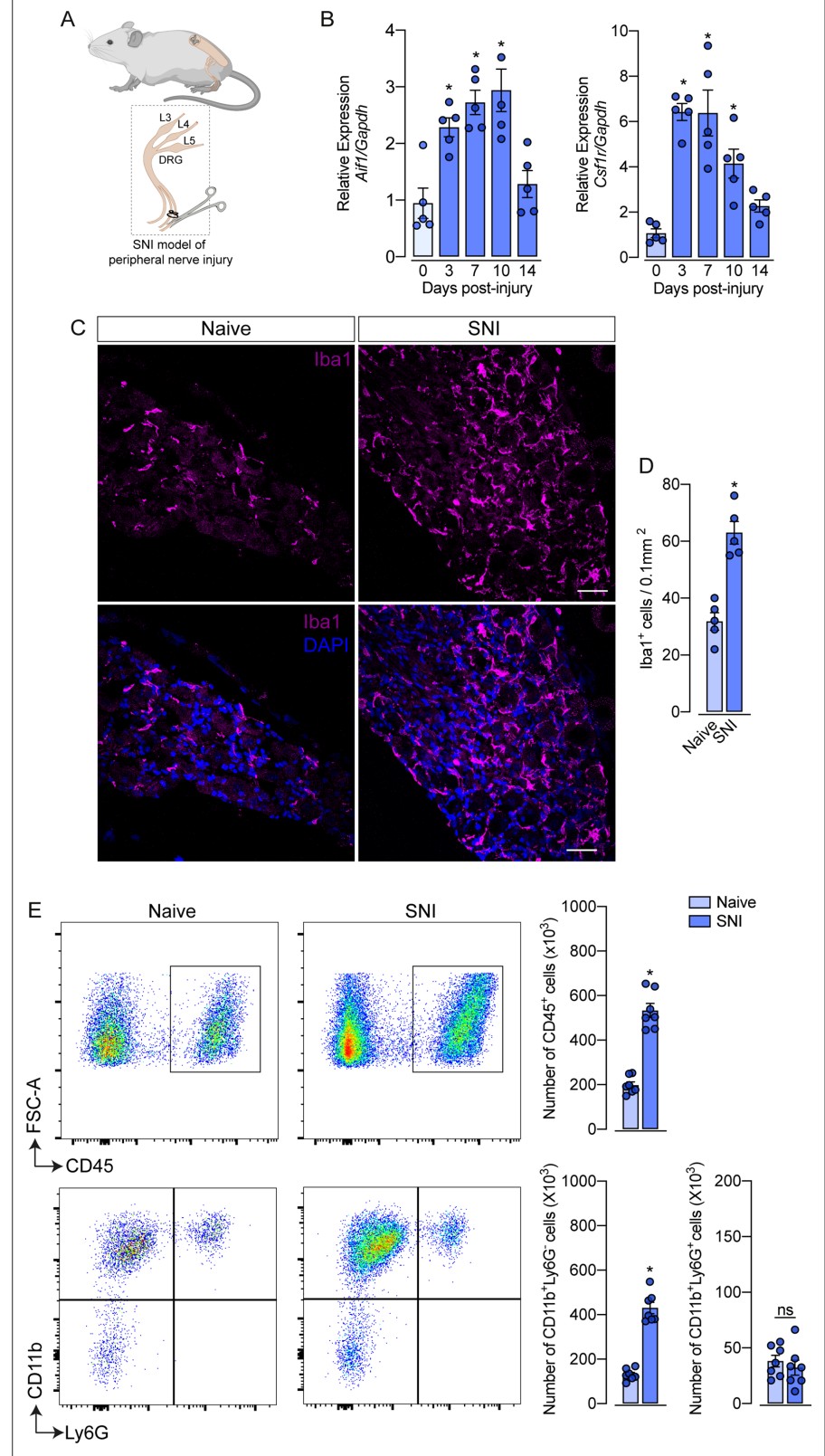

**Figure 1.** Spared nerve injury (SNI) model induced an increase in the number of macrophages in the dorsal root ganglia (DRG). (**A**) Schematic representation of the experimental design of SNI induction in mice, showing the sciatic nerve, its branches and the dorsal root ganglia (DRG; L3, L4 and L5) harvested. (**B**) Time course of *Csf1r* and *Aif1* mRNA expression relative to *Gapdh* in the DRGs from naïve WT mice (day 0) or 3, 7, 10, and 14 days post-

*Figure 1 continued on next page*

*Figure 1 continued*

injury (n=4–5). (**C**) Representative confocal images of L4 DRG from WT mice at 7 days after SNI. Scale bars: 50 µm. (**D**) Quantification of macrophages (Iba1 + cells) in DRGs at 7 days after SNI (n=5). (**E**) Representative dot plots and absolute number of CD45$^+$, CD11b$^+$Ly6G$^-$, and CD11b$^+$Ly6G$^+$ cells in the DRGs (L3–L5) at 7 days after SNI by flow cytometry (n=7). Results are shown as the mean ± SEM. p-values were determined by one-way ANOVA followed by Bonferroni's post hoc test. *p<0.05; ns, not significant. Data are representative of at least three independent experiments.

up-regulated from day 3, reaching a peak between 7 and 10 days after SNI (*Figure 1B*). Additionally, immunofluorescence analysis revealed an increased number of macrophages (Iba1$^+$ cells) in the DRGs at 7 days after nerve injury (*Figure 1C and D*). Using flow cytometry, we confirmed an increase in the number of macrophages (CD11b$^+$ Ly6G$^-$ cells) in the DRGs at 7 days after SNI (*Figure 1E*), while no significant changes were observed in the number of neutrophils (CD11b$^+$ Ly6G$^+$ cells). Together, these data confirmed that after peripheral nerve injury, the macrophage population increases in the sensory ganglia.

## CX3CR1$^+$ macrophage population expands in the DRGs after peripheral nerve injury independently of CCR2$^+$ monocytes

As mentioned previously, it is still unclear whether circulating monocytes can infiltrate into the sensory ganglia (e.g. DRG) and might account for the increase in the number of macrophages observed after peripheral nerve injury. To address this question, we employed different experimental approaches. Firstly, we used *Cx3cr1*$^{GFP/+}$/*Ccr2*$^{RFP/+}$ reporter mice, which might be useful for distinguishing the typical CX3CR1$^+$ tissue-resident macrophages from CCR2$^+$ blood monocytes (*Jung et al., 2000*; *Saederup et al., 2010*). In naive conditions, CX3CR1$^+$ cells predominate in the sensory ganglia compared to CCR2$^+$ cells (*Figure 2A and B*). Noteworthy, we found that ~97% of CX3CR1$^+$ cells in the DRGs are Iba1$^+$ and vice-versa (*Figure 2—figure supplement 1A*), indicating that all resident macrophages in the sensory ganglia are CX3CR1$^+$.

Notably, while the number of CX3CR1$^+$ cells in the DRGs increased after SNI (*Figure 2B*), CCR2$^+$ inflammatory monocytes numbers did not change (*Figure 2B*), indicating that the circulating CCR2$^+$ monocytes did not infiltrate into the sensory ganglia after peripheral nerve injury. As a positive control of CCR2$^+$ monocyte infiltration into the DRGs, we used a murine model of HSV-1 infection (*Silva et al., 2017*). As expected, we observed a significant CCR2$^+$ blood-monocyte infiltration in the DRGs of HSV-1-infected mice (*Figure 2—figure supplement 2A* and B). Corroborating these data, the number of CD11b$^+$ Ly6C$^+$ Ly6G$^-$ cells (peripheral monocytes) did not change in the DRGs after SNI compared to naive animals (*Figure 2—figure supplement 3A*). In addition, we also found an increase in the expression of *Cx3cr1* in the DRGs after SNI, and flow cytometry analysis confirmed a significant increase in the number of CD11b$^+$ CX3CR1$^+$ cells when compared to the naive group (*Figure 2C and D*).

Additionally, using deficient mice for the chemokine receptor CCR2 (*Ccr2*$^{-/-}$ mice), we noticed that the increase of *Cx3cr1* gene expression after SNI did not change in these mice compared to *Ccr2*-sufficient animals (*Figure 3A*). Furthermore, flow cytometry (CD11b$^+$ CX3CR1$^+$ cells) and immunofluorescence (Iba1$^+$ cells) analyses showed a similar number of macrophages in the DRGs from *Ccr2*$^{-/-}$ mice compared to WT mice in a naive condition and increased at the same level after SNI (*Figure 3B and D*). These results suggest that CCR2$^+$ monocyte infiltration is not required to expand CX3CR1$^+$ resident macrophages in the DRGs after peripheral nerve injury.

To support these data, we performed parabiosis as an additional approach to elucidate the possible infiltration of blood-borne monocytes in the DRG after SNI. Pairs of sex- and weight-matched C57BL/6-Tg (CAG-EGFP) mice and wild-type mice were surgically joined and remained to share circulation for 4 weeks, followed by the SNI (*Figure 4A and B*). In agreement with our earlier results, the number of macrophages (Iba1$^+$ cells) increased in the DRGs of WT mice 7 days after SNI. However, the number of GFP$^+$ cells in the DRGs of WT mice remained constant in both naïve and SNI groups (*Figure 4C and D*). Next, using the same experimental approach, we performed parabiosis in WT and *Cx3cr1*$^{GFP/+}$/*Ccr2*$^{RFP/+}$ mice (*Figure 4—figure supplement 1A and B*). Despite the increase of macrophages (Iba1$^+$ cells) in the DRGs after SNI, we could not detect substantial numbers of CCR2$^+$ or CX3CR1$^+$ cells in SNI compared to naive mice (*Figure 4—figure supplement 1C and D*). Finally, we used the *Ms4a3*$^{Cre}$ fate-mapping model to identify monocyte and monocyte-derived lineage cells

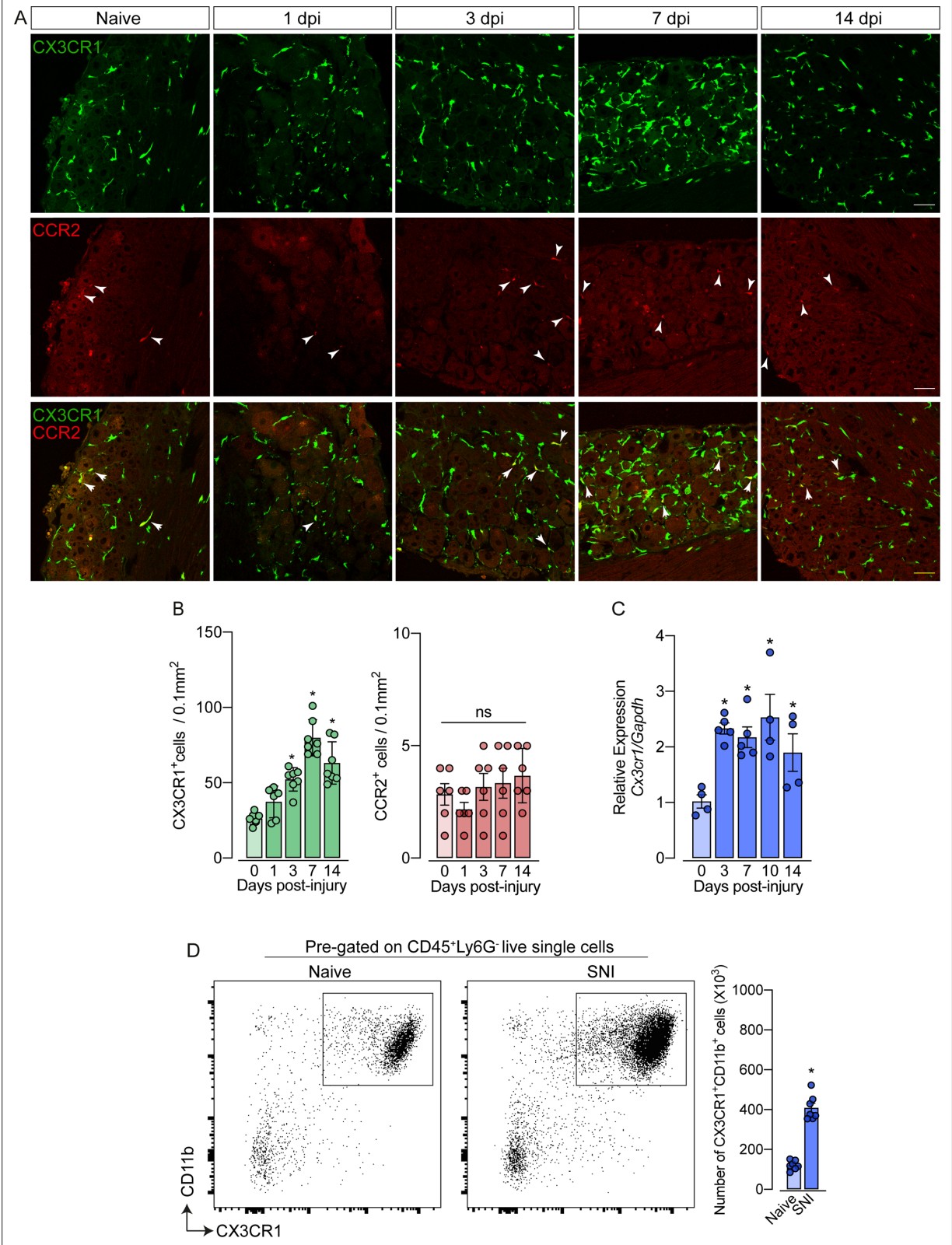

**Figure 2.** Spared nerve injury (SNI) induced an increase of CX3CR1+ macrophages, but not of CCR2+ monocytes in the dorsal root ganglias (DRGs). (**A**) Representative confocal images of L4 DRG from naive *Cx3cr1^{GFP/+}/Ccr2^{RFP/+}* mice or 1, 3, 7, and 14 days post-injury (dpi). CX3CR1-GFP+macrophages are shown in green and CCR2-RFP+monocytes are shown in red (indicated with white arrows). Scale bars: 50 μm. (**B**) Quantification of macrophages (CX3CR1-GFP+ cells) and monocytes (CCR2-RFP+ cells) in DRGs naive (day 0) or 1, 3, 7, and 14 days post-injury (n=6–7). (**C**) Time course of *Cx3cr1* mRNA

*Figure 2 continued on next page*

*Figure 2 continued*

expression relative to *Gapdh* in the DRGs from naïve WT mice (day 0) or after 3, 7, 10, and 14 days post-injury (n=4–5). (**D**) Representative dot plots and absolute number of CX3CR1+CD11b+ cells in the DRGs (L3–L5) from *Cx3cr1*$^{GFP/+}$ mice at 7 days after SNI by flow cytometry (n=7). Dots represent individual mice. Results are shown as the mean ± SEM. p-values were determined by (**A–C**) one-way ANOVA followed by Bonferroni's post hoc test and (**D**) two-tailed Student's *t*-test. *, p<0.05; ns, not significant. Data are representative of at least three independent experiments.

The online version of this article includes the following figure supplement(s) for figure 2:

**Figure supplement 1.** CX3CR1+ and Iba1+ cells increased in dorsal root ganglias (DRGs) after spared nerve injury (SNI).

**Figure supplement 2.** CCR2+ monocytes increase in the dorsal root ganglias (DRGs) after HSV-1 peripheral infection.

**Figure supplement 3.** There is no increase in the number of inflammatory monocytes in the dorsal root ganglias (DRGs) after spared nerve injury (SNI).

(*Liu et al., 2019*). We found that, in naive condition, the number of *Ms4a3*$^{Cre-tdTomato}$/Iba1+ cells are relatively low in the DRGs (~17%), indicating a small contribution of hematopoietic-derived monocytes to the pool of resident macrophages in the DRGs (*Figure 5A*). In addition, the number of monocytes or monocyte-derived cells in the DRGs did not change significantly after SNI (*Figure 5A and B*), further indicating that blood monocytes are not infiltrating the DRGs after SNI. Since the *Ms4a3*$^{Cre-tdTomato}$ model also allows the tracing of granulocytes (*Liu et al., 2019*), these data also indicated that neutrophils did not infiltrate the DRGs after peripheral nerve injury. Altogether, these findings strongly suggest that the increase in the number of macrophages in the DRGs after peripheral nerve injury is likely to result from the expansion/proliferation of CX3CR1+ resident macrophages regardless of peripheral blood CCR2+ monocytes infiltration.

## CX3CR1+ resident macrophages proliferate in the sensory ganglia which is associated with neuropathic pain development

Given the increase in the number of macrophages in the DRGs after SNI in the absence of circulating monocyte infiltration, we hypothesized that CX3CR1+ resident macrophages in the DRG could undergo rapid and local proliferation after peripheral nerve injury. Corroborating this hypothesis, we found that the expansion of CX3CR1+ macrophages in the DRGs was accompanied by an increase in Ki67+ staining in CX3CR1+ macrophages, three days after SNI indicating a high proliferation profile (*Figure 6A and B*). Notably, 7 days after injury, a significant number of cells are still proliferating, however, few of them are CX3CR1+ macrophages. This indicates that other cell populations (e.g. satellite glial cells, Schwann cells, or mesenchymal cells) also undergo proliferation after SNI. Noteworthy, the increase in the number of Ki67+ macrophages in the DRGs from *Ccr2*$^{-/-}$ mice after SNI was similar to the numbers observed in the DRGs from WT mice (*Figure 6—figure supplement 1A and B*).

The fact that resident sNAMs in the sensory ganglia are involved in the development of neuropathic pain (*Yu et al., 2020*), prompted us to test the hypothesis that inhibition of proliferation of these macrophages would inhibit the development of pain state. In fact, the intrathecal treatment (which targets cells in the DRGs; *Wang et al., 2005*) of mice with AraC, a cell proliferation inhibitor, reduced SNI-induced mechanical pain hypersensitivity (mechanical allodynia) that was associated with decreased numbers of CX3CR1+ cells in the DRGs (*Figure 6C and D*). Noteworthy, this dose of AraC injected intrathecally did not change the baseline of the mechanical nociceptive threshold (*Gu et al., 2016*). Altogether, these results indicate that the CX3CR1+ resident macrophage population is expanding, accounting for the increase in the number of macrophages in the sensory ganglia and neuropathic pain (e.g. mechanical allodynia) development observed after peripheral nerve injury.

## CX3CR1 signaling mediates sNAMs expansion in the DRGs and neuropathic pain development after peripheral nerve injury

After characterizing the resident macrophages proliferation in the sensory ganglia triggered by peripheral nerve injury, we sought to investigate possible mechanisms involved in this process, since CX3CR1 is expressed in sNAMs (*Wang et al., 2020*; *Kolter et al., 2019*; *Chakarov et al., 2019*) and there is evidence that CX3CR1 signaling mediates microglia proliferation in the spinal cord after peripheral nerve injury (*Gu et al., 2016*; *Suter et al., 2009*; *Rotterman and Alvarez, 2020*). Herein, we also observed that the expression of *Cx3cl1*, the CX3CR1 ligand, increased in the DRGs after SNI (*Figure 7A*). Thus, we tested whether CX3CR1 signaling would be involved in SNI-induced sNAMs expansion. For this purpose, we took advantage of *Cx3cr1*$^{GFP/GFP}$ mice (*Cx3cr1 null mice*)

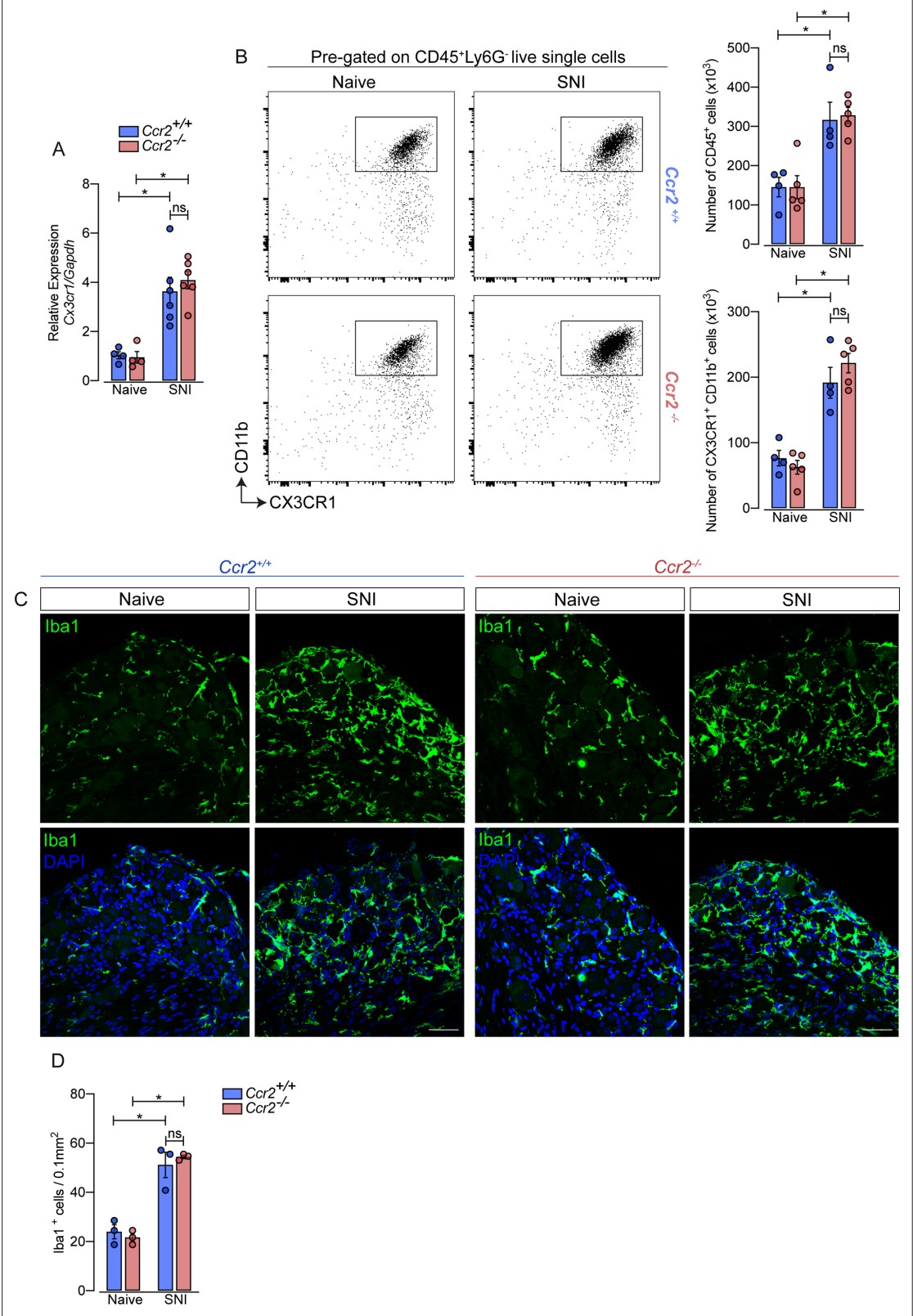

**Figure 3.** Macrophages increased in the sensory ganglia after spared nerve injury (SNI) independent of CCR2 signaling. (**A**) RT-qPCR analysis of *Cx3cr1* mRNA expression relative to *Gapdh* in the DRGs from *Ccr2^-/-* and *Ccr2^+/+* mice after 7 days of SNI (n=4–6). (**B**) Representative dot plots and absolute number of CD45^+ and CX3CR1^+CD11b^+ cells in the dorsal root ganglias (DRGs) (L3–L5) from *Ccr2^-/-* and *Ccr2^+/+* mice 7 days after SNI by flow cytometry (n=4–5). (**C**) Representative confocal images of L4 DRG from *Ccr2^-/-* and *Ccr2^+/+* mice 7 days after SNI. Scale bars: 50 μm. (**D**) Quantification

*Figure 3 continued on next page*

*Figure 3 continued*

of macrophages (Iba1+ cells) in DRGs at 7 days after SNI (n=3). Results are shown as the mean ± SEM. p-values were determined by one-way ANOVA followed by Bonferroni's post hoc test. *p<0.05; ns, not significant. Data are representative of at least two independent experiments.

and evaluated the expansion of macrophages in the DRGs after SNI compared to their littermates (*Cx3cr1*$^{GFP/+}$ mice). Notably, *Cx3cr1*$^{GFP/GFP}$ mice in naive condition did not show any difference in the number of CX3CR1$^+$CD11b$^+$ cells in the sensory ganglia (DRGs L3-L5) compared to heterozygous littermate controls (*Cx3cr1*$^{GFP/+}$ mice) (*Figure 7B*). Nevertheless, the increase of the CX3CR1$^+$CD11b$^+$ cells population observed after SNI in the DRGs was reduced in the absence of the CX3CR1 signaling (*Figure 7B*). These flow cytometry data were confirmed by immunofluorescence analysis that showed no difference, at the naive condition in the number of CX3CR1$^+$ Iba1$^+$ macrophages in the DRGs from *Cx3cr1*$^{GFP/GFP}$ mice compared to DRGs from *Cx3cr1*$^{GFP/+}$ mice (*Figure 7C and D*). Furthermore, the number of CX3CR1$^+$ Iba1$^+$ macrophages after SNI in the DRGs from *Cx3cr1*$^{GFP/GFP}$ mice was reduced when compared to the DRGs from *Cx3cr1*$^{GFP/+}$ mice (*Figure 7C and D*).

Supporting the involvement of CX3CR1 signaling on sNAMs proliferation in the DRGs, we found that the number of macrophages proliferating (CX3CR1$^+$ Ki67$^+$ cells) after SNI in the DRGs from *Cx3cr1*$^{GFP/GFP}$ mice was reduced when compared to the DRGs from *Cx3cr1*$^{GFP/+}$ mice (*Figure 7—figure supplement 1A and B*). Finally, we also found that SNI-mechanical pain hypersensitivity was also reduced in *Cx3cr1*$^{GFP/GFP}$ mice compared to *Cx3cr1*$^{GFP/+}$ mice (*Figure 7E*). These results suggest that CX3CR1 signaling does not control the survival/seeding of sNAMs in the sensory ganglia, but might be important for their proliferation and consequently to neuropathic pain development after peripheral nerve injury.

## sNAMs are the main source of pro-inflammatory cytokines production after peripheral nerve injury: role of CX3CR1 signaling

After peripheral nerve injury, the expression of several cytokines and chemokines increases in the sensory ganglia. Among these cytokines, TNF, IL-1b, and IL-6 seem to be the most important (*Yu et al., 2020*; *Kwon et al., 2013*; *Ji et al., 2014*). However, the cellular source of these cytokines in the sensory ganglia after peripheral nerve injury is still controversial. Herein, we confirmed this evidence and observed an up-regulation of *Tnf*, *Il1b*, and *Il6* transcripts in the DRGs of mice 7 days after SNI compared to naive-control mice (*Figure 8A*). In an attempt to identify the cellular source of these cytokines, we initially took advantage of publicly available single-cell RNAseq data from DRGs cells (*Avraham et al., 2020*). After the re-analyze of these data, we were able to identify 12 different cellular clusters, including sNAMs (*Figure 8B*). In addition, the expression of *Tnf and Il1b* after peripheral nerve injury was confined in the sNAMs cluster (*Figure 8B*), while *Il6* expression was not conclusively defined (*Figure 8B*). To confirm these data, we next performed cell sorting and subsequent qPCR analysis of both CD45$^-$ and CX3CR1$^+$CD11b$^+$ cells from DRGs collected from *Cx3cr1*$^{GFP/+}$ mice ipsilateral and contralateral to the SNI injury. We found that *Tnf and Il1b* transcripts are detected substantially in CX3CR1$^+$CD11b$^+$ cells harvested from DRGs, and their expression increased only in this specific cell population after SNI (*Figure 8C*). On the other hand, *Il6* expression was detected mainly on CD45$^-$ cells (*Figure 8C*). Finally, we sought to investigate whether the production of pro-inflammatory cytokines after peripheral nerve injury is also dependent on CX3CR1 signaling. While we detected an increase of *Tnf and Il1b* in the DRGs from WT mice after SNI, these genes were reduced in DRGs from *Cx3cr1*$^{GFP/GFP}$ mice (*Figure 8D*). On the other hand, *Il6* expression increased in the DRGs similarly in both mice genotypes (*Figure 8D*). Altogether, these results indicated that sNAMs are the main source of CX3CR1 signaling-dependent pro-inflammatory cytokines (e.g. IL-b and TNF) in the sensory ganglia after peripheral nerve injury, whereas IL-6 seems to be induced mainly in non-immune cells.

## Discussion

Tissue-resident macrophages are abundant in all tissues, whereby they contribute to the maintenance of tissue homeostasis and act as effectors of innate immunity. Among the subtypes of tissue-resident macrophages, sNAMs are distributed along the sciatic nerve and closer to the cell body of primary sensory neurons located in the sensory ganglia, where they might be involved in physiological and

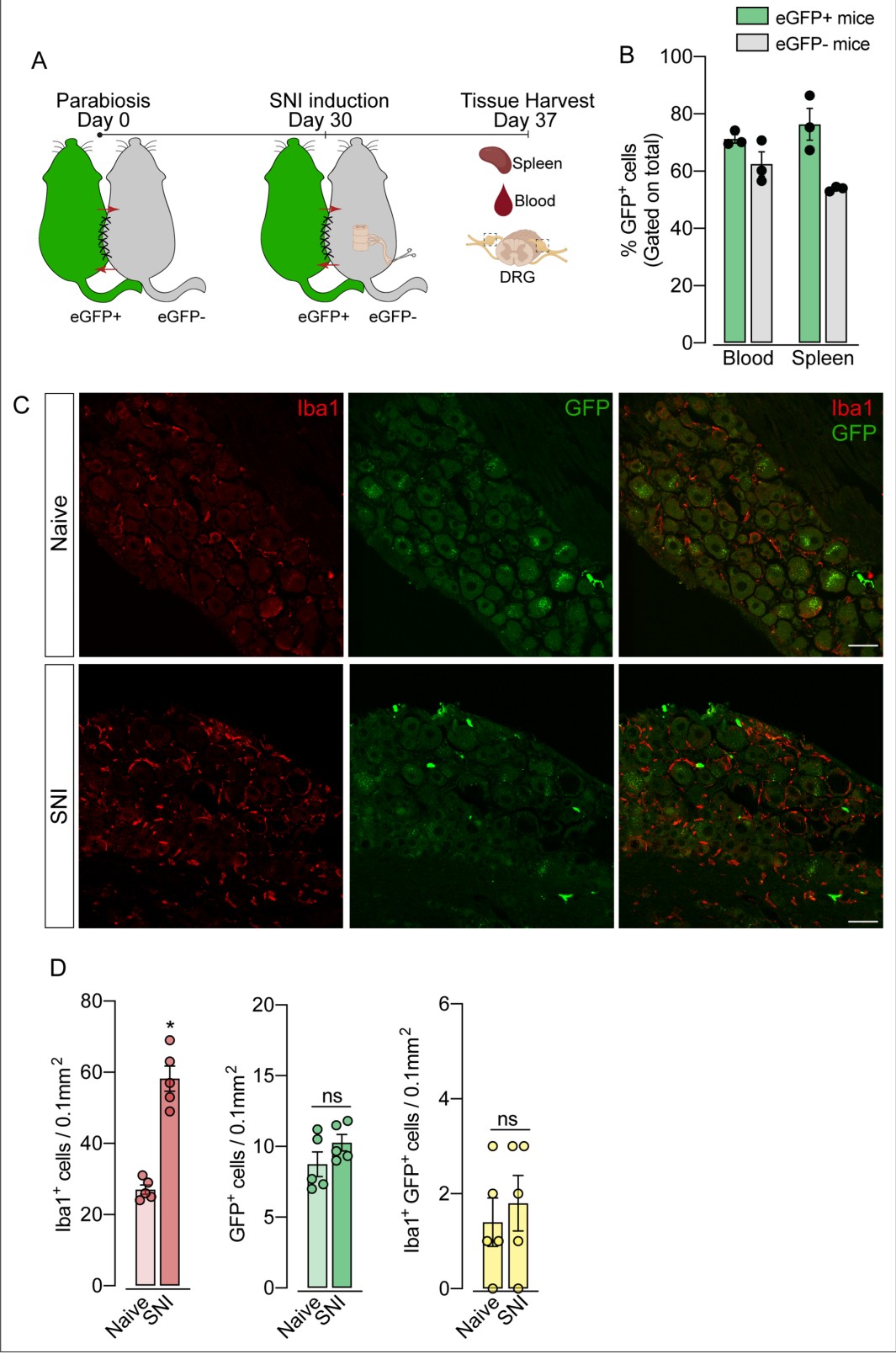

**Figure 4.** Blood leukocytes did not infiltrate the dorsal root ganglias (DRGs) after spared nerve injury (SNI). (**A**) Schematic representation of parabiotic mouse pairs: eGFP+ (C57BL/6-(Tg[CAG-EGFP])) and eGFP– (C57BL/6 J) mice. After 30 days, SNI was induced in eGFP– mice and maintained for 7 days, then spleen, blood, and DRGs were harvested. (**B**) Frequency of GFP+ cells in the blood and spleen from eGFP+ and eGFP– mice after 37 days of

*Figure 4 continued on next page*

*Figure 4 continued*

parabiosis. (**C**) Representative confocal images of L4 DRG from naive or SNI eGFP⁻ parabionts mice. GFP⁺ cells are shown in green and Iba1⁺ cells are shown in red. Scale bars: 50 μm. (**D**) Quantification of GFP⁺, Iba1⁺, and GFP⁺ Iba1⁺ cells in DRGs naive or ipsilateral (SNI) (n=5 pairs of mice). Data are representative of two independent experiments. Results are shown as the mean ± SEM. p-values were determined by two-tailed Student's *t*-test. *, p<0.05; ns, not significant. Data are representative of at least two independent experiments.

The online version of this article includes the following figure supplement(s) for figure 4:

**Figure supplement 1.** Parabiosis reveals no infiltration of circulating CX3CR1⁺ or CCR2⁺ cells in the dorsal root ganglia (DRG) after spared nerve injury (SNI).

---

pathophysiological processes (***Wang et al., 2020***; ***Yu et al., 2020***; ***Montague et al., 2018***; ***Kalinski et al., 2020***; ***Old et al., 2014***; ***Ydens et al., 2020***). Evidence indicates that after peripheral nerve injury macrophage population increases in the sensory ganglia and accounts for neuropathic pain development and neuroregeneration, through the production of pronociceptive mediators, such as cytokines and chemokines (***Yu et al., 2020***; ***Guimarães et al., 2019***). The nerve injury-induced sNAMs expansion in the sensory ganglia has been ascribed mainly to the infiltration of blood monocytes (***Kim et al., 2011c***). Nevertheless, recent evidence suggests that sensory ganglia resident sNAMs also proliferate after peripheral nerve injury (***Krishnan et al., 2018***). Thus, additional data are necessary to clarify the mechanisms involved in the increase of macrophages in the sensory ganglia after peripheral nerve injury.

In this study, using different experimental approaches, we demonstrated that the increase of macrophages in the DRG after SNI is mainly due to the proliferation of CX3CR1⁺ residents and does not rely on the infiltration of blood monocytes. The proliferation of these sensory ganglia sNAMs seems to be involved in the development of mechanical pain hypersensitivity (neuropathic pain) triggered by peripheral nerve injury. We also established that CX3CR1 signaling on sNAMs causes their propagation and activation, stimulating the production of TNF and IL-1b cytokines. It was long considered that after peripheral nerve injury, the production of inflammatory mediators in the DRG is associated with the infiltration of peripheral blood monocytes, assumed from the increased expression of some markers such as IBA1, CD11b, ED1, CD68, and/or F4/80 (***Liu et al., 2010***; ***Kim et al., 2011b***; ***Huang et al., 2014***; ***Kallenborn-Gerhardt et al., 2014***; ***Luo et al., 2019***). In order to dispute this hypothesis, we initially characterized DRG macrophages based on the expression of CCR2 and CX3CR1 receptors to distinguish two major populations of resident macrophages or peripheral blood monocytes, respectively (***Gordon and Taylor, 2005***).

We took advantage of the transgenic reporter mice *Cx3cr1*^GFP/+^/*Ccr2*^RFP/+^ to trace tissue-resident macrophages (CX3CR1⁺) and monocytes recruited from the blood to inflamed tissue (CCR2⁺) after peripheral nerve injury. The same strategy has been used to distinguish peripheral monocyte infiltration in the central nervous system and resident microglia in different disease models (***Mizutani et al., 2012***; ***Yamasaki et al., 2014***; ***Chen et al., 2020***). Contrary to previous studies, the number of CCR2⁺ monocytes remained constant in the DRGs after SNI, despite the significant increase in the number of CX3CR1⁺ resident macrophages (***Illias et al., 2018***; ***Zhang et al., 2021***; ***Kim et al., 2011c***). These results initially could indicate that blood CCR2⁺ monocytes fail to infiltrate the sensory ganglia after peripheral nerve injury. However, because these mice are knock-in reporters and the expression of CCR2 could be downregulated during inflammatory conditions (***Cebinelli et al., 2021***), additional strategies would be necessary. Furthermore, a population of blood monocytes (the patrolling monocytes) also down-regulates CCR2 and starts to express CX3CR1 (***Ginhoux and Jung, 2014***), and these patrolling monocytes could be also infiltrating the DRGs after SNI. Our parabiosis experiments support the idea that blood-derived monocytes are not significantly recruited and, neither leukocyte subtypes can infiltrate the sensory ganglia after peripheral nerve injury. The fail of leukocyte infiltration has been also demonstrated in the spinal cord after peripheral nerve injury (***Gu et al., 2016***; ***Santa-Cecília et al., 2019***). Even though the peripheral nerve injury did not trigger an infiltration of circulating monocytes into the DRG, few of these cells were observed in the tissue parenchyma under homeostasis. These data are in line with recent findings demonstrating that BM-derived monocytes may contribute to a modest subset of PNS macrophages at steady-state (***Feng et al., 2023***). Additionally, the maintenance of DRG-resident macrophages can be slowly replaced by peripheral monocytes, while the majority of sNAMs arise from embryonic precursors and must be able to proliferate and

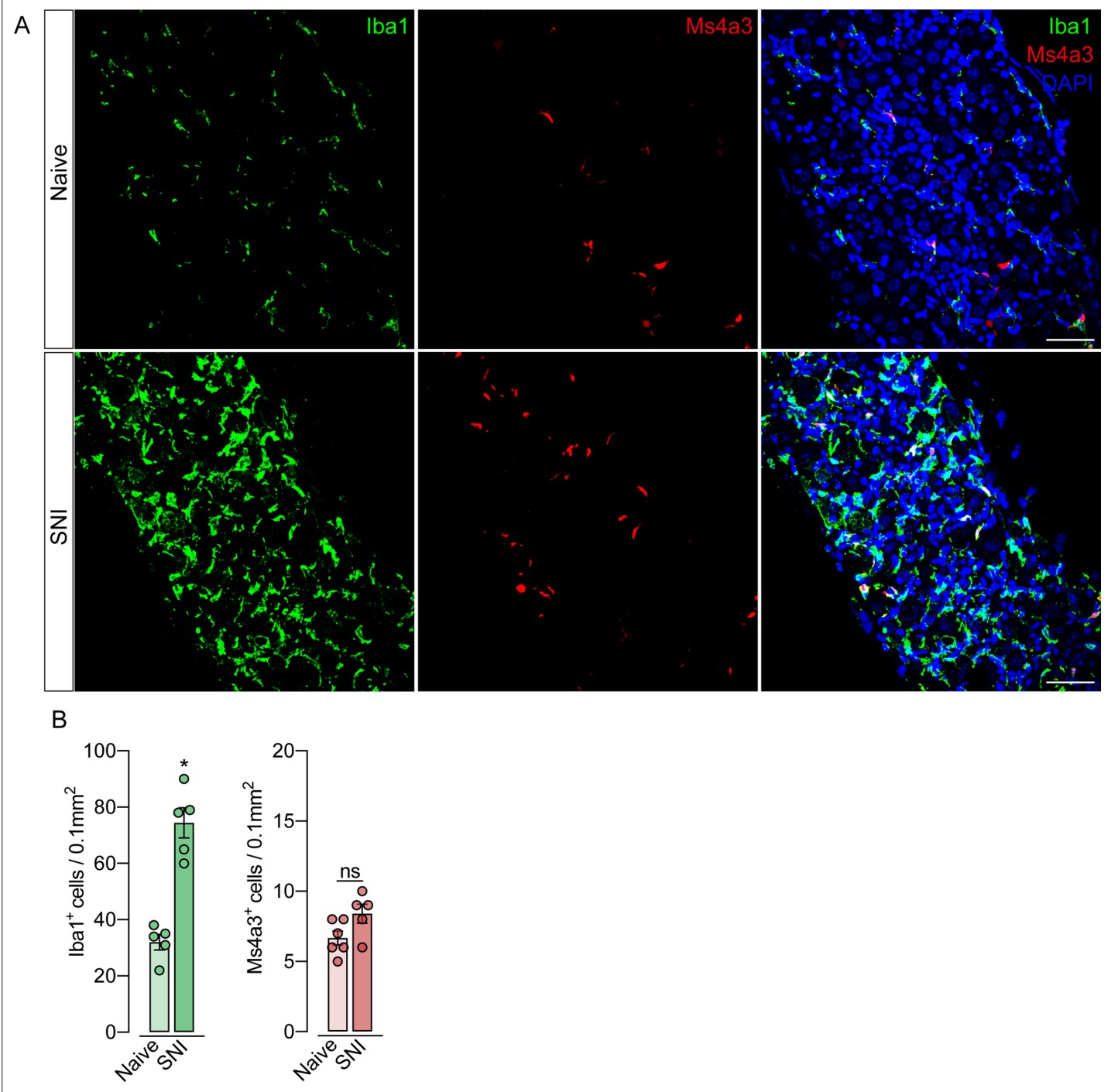

**Figure 5.** Monocytes or monocyte-derived cells did not infiltrate the dorsal root ganglias (DRGs) after spared nerve injury (SNI). (**A**) Representative confocal images of L4 DRG from *Ms4a3^Cre-tdTomato* mice after 7 days of SNI. Iba1$^+$ cells are shown in green and Ms4a3$^+$ cells are shown in red. Scale bars: 50 µm. (**B**) Quantification of Iba1$^+$ and Ms4a3$^+$ cells in DRGs 7 days after SNI (n=5–6). Results are shown as the mean ± SEM. p-values were determined by two-tailed Student's *t*-test. *p<0.05; ns, not significant. Data are representative of at least two independent experiments.

self-renew (***Feng et al., 2023***). This hypothesis was also confirmed by our data using the *Ms4a3^Cre-tdTomato* fate-mapping model that further allowed us to track monocytes and monocyte-derived cells (***Liu et al., 2019***). Using this strategy, we also found a very small proportion of macrophages in the sensory ganglia originate from hematopoietic derived-monocytes. Additionally, these data also indicate that these macrophages (derived from monocytes, *Ms4a4^tdTomato*) did not expand significantly

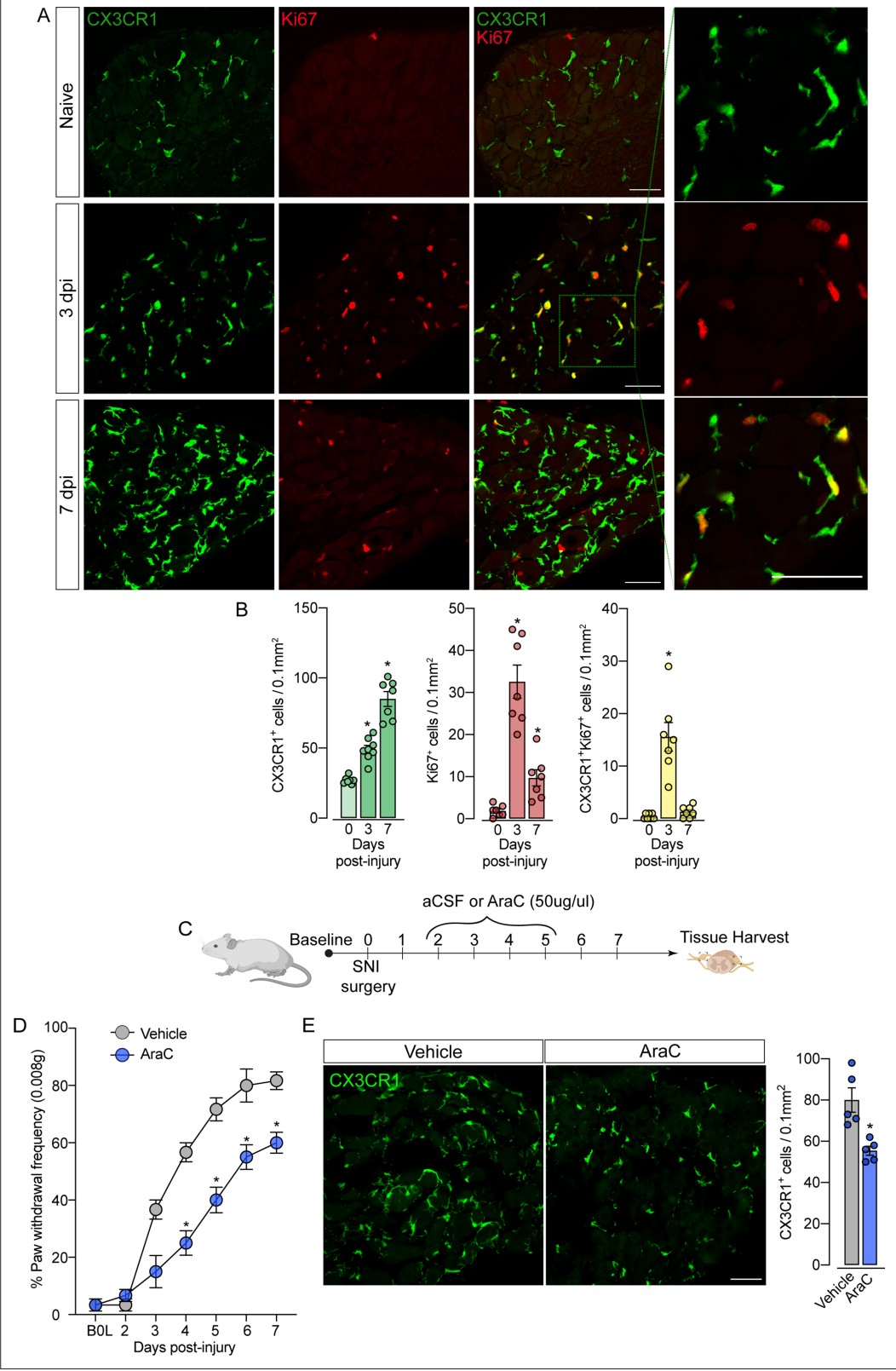

**Figure 6.** Sensory neuron-associated macrophages (sNAMs) proliferate in the sensory ganglia after peripheral nerve injury: role in mechanical allodynia. (**A**) Representative confocal images of L4 DRG from naive *Cx3cr1GFP/+* mice or 3 and 7 days post-injury (dpi). CX3CR1+ cells are shown in green and Ki67+ cells are shown in red. Dotted boxes show regions of higher magnification in the dorsal root ganglia (DRG). Scale bars: 50 μm.

*Figure 6 continued on next page*

*Figure 6 continued*

(**B**) Quantification of CX3CR1$^+$, Ki67$^+$, and CX3CR1$^+$Ki67$^+$ cells in DRGs naive (day 0) or 3 and 7 dpi (n=7–8). (**C**) Schematic representation of intrathecal treatment in *Cx3cr1*$^{GFP/+}$ mice with AraC (Cytarabine) or aCSF (vehicle) for 4 consecutive days starting 2 days after spared nerve injury (SNI) induction. 7 days post-injury the L4 DRGs were harvested. (**D**) Mechanical nociception was evaluated (for 7 days) by paw withdrawal frequency using 0.008 g of von Frey filament. (**E**) Representative confocal images of L4 DRG from *Cx3cr1*$^{GFP/+}$ mice after AraC or aCSF treatment. CX3CR1$^+$ cells are shown in green. Scale bars: 50 μm. Quantification of CX3CR1$^+$ cells in DRGs after injections (n=5). Results are shown as the mean ± SEM. p-values were determined by one-way ANOVA followed by Bonferroni's post hoc test. *p<0.05. Data are representative of at least three independent experiments.

The online version of this article includes the following figure supplement(s) for figure 6:

**Figure supplement 1.** Macrophage proliferation in the sensory ganglia did not change in the absence of CCR2 signaling.

after peripheral nerve injury. Notably, our data obtained using *Ms4a3*$^{Cre-tdTomato}$ fate-mapping model also confirmed that neutrophils are not able to infiltrate the sensory ganglia after peripheral nerve injury, which is also a controversy in the literature (*Morin et al., 2007*).

An important question that arises from these data is why peripheral blood-circulating cells are not able to infiltrate into the DRG after peripheral nerve injury assuming that there is a significant production of proinflammatory mediators in the tissue. Although the reasons are not immediately apparent, the possible explanation is the blood dorsal root ganglion barrier, formed by the perineurium and endoneurial blood vessels. This barrier protects and maintains the PNS in an appropriate physicochemical environment (*Reinhold and Rittner, 2020*). The perineurium is a thick layer of connective tissue whose cells have a non-polarized architecture and are interconnected by tight junctions (TJ), gap junctions, and adherens junctions (AJ), similar to the composition of the CNS blood-brain barrier. The endoneurial vessels are composed of a network of arterioles, venules, and non-fenestrated capillaries. The endothelial cells that create this vascular network are also sealed by TJ, but more permeable than the perineurium, since there must be a controlled exchange between blood and nerve to allow neural nutrition (*Reinhold and Rittner, 2017*). While these barriers may limit the infiltration of circulating monocytes into the DRG under homeostasis, it remains unclear how peripheral nerve damage induced by SNI can affect its integrity. Previous studies suggest that after crush injury, there may be a loss and recovery of the blood dorsal root ganglion barrier junction, associated with the expression of intercellular junctional proteins (*Hirakawa et al., 2003*). A recent study shows that CD8$^+$ T cells, that did not infiltrate the sensory ganglia of adult mice after peripheral nerve injury, are able to infiltrate when the injury is performed in 2 years old mice (*Zhou et al., 2022*). These data indicate that the process of aging might alter the sensory ganglia-blood barrier. In addition, although immune cells seem to be unable to infiltrate the cell-body-rich area of the DRGs after peripheral nerve injury (at least in young mice), there is recent evidence of accumulation of leukocytes into the dorsal root leptomeninges that cover the sensory ganglia (*Du et al., 2018*; *Maganin et al., 2022*). Nevertheless, additional studies will be necessary to elucidate leukocyte trafficking into these regions after peripheral nerve injury.

Our study further showed that the SNI-induced increase of sNAMs in the sensory ganglia is a consequence of the local proliferation of CX3CR1-resident macrophages. Our data showing an increase in the expression of Ki67, a classical marker of cell proliferation, in CX3CR1$^+$ cells before the expansion of this population, is an important finding to support this conclusion. Additionally, a recent study that analyzed the single-cell transcriptome of DRG cells, after peripheral nerve injury, also found an increase in genes related to proliferation in the macrophage population (*Avraham et al., 2020*). Herein, we also provided evidence that the proliferation of these sensory ganglia sNAMs might mediate the development of neuropathic pain, suggesting that targeting this process would be an interesting approach to prevent the neuropathic pain. Noteworthy, we cannot exclude that the current approach we have used to target sNAMs proliferation (intrathecal injection of AraC) is also inhibiting spinal cord microgliosis (*Gu et al., 2016*).

It is still unclear how the peripheral nerve injury leads to the distal proliferation/activation of sNAMs seeded in the DRG. Some studies have indicated that constant activation of neurons following peripheral nerve injury results in CX3CL1 production in the spinal cord, which in turn induces activation/proliferation of local microglia (*Verge et al., 2004*; *Peng et al., 2016*; *Clark et al., 2007*). Here, we

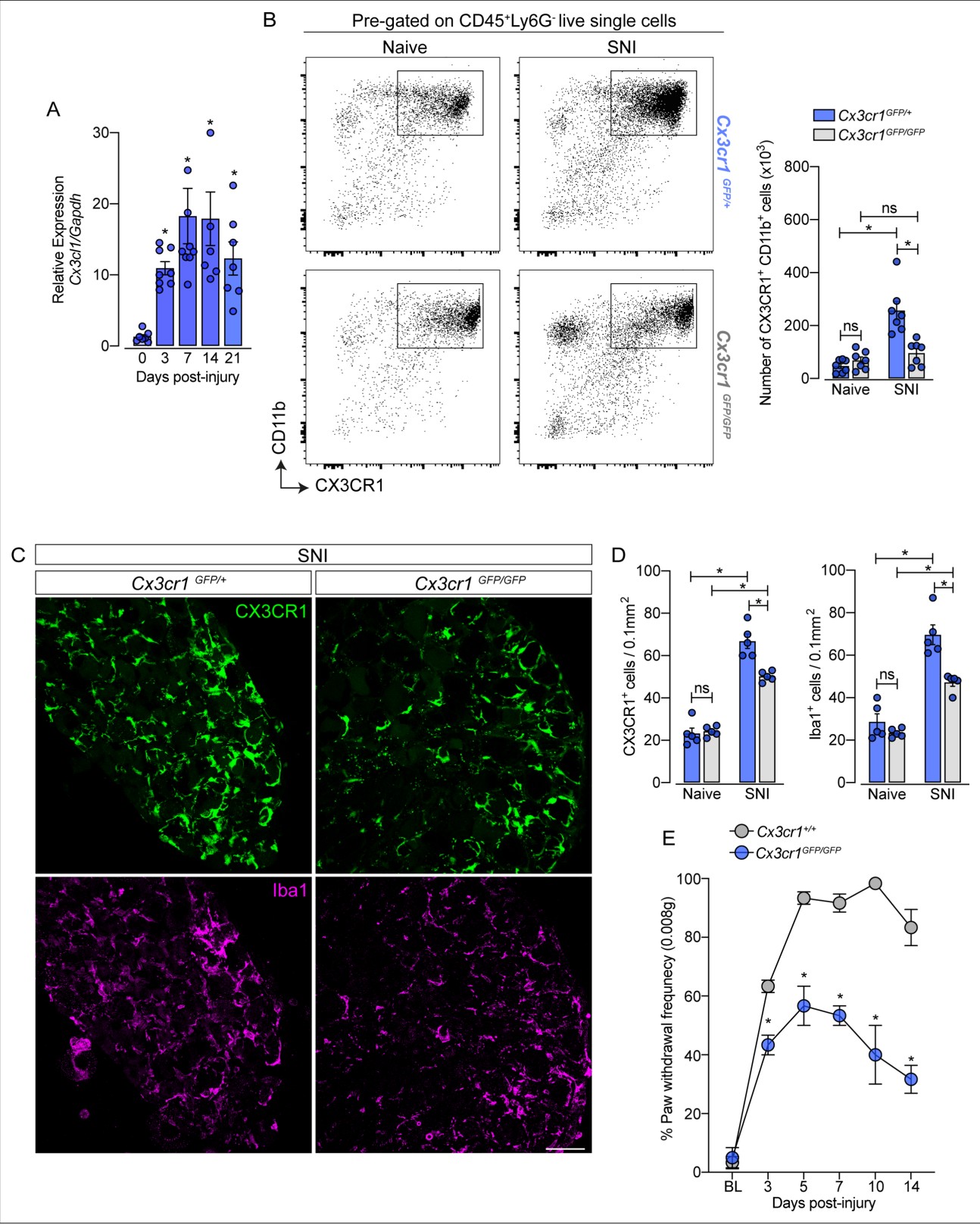

**Figure 7.** CX3CR1 signaling is involved in macrophage expansion in the dorsal root ganglia (DRG) after peripheral nerve injury. (**A**) Time course of *Cx3cl1* mRNA expression relative to *Gapdh* in the DRGs from naive (day 0) or after 3, 7, 14, and 21 days post-injury (n=7–8). (**B**) Representative dot plots and absolute number of CX3CR1+CD11b+ cells in the DRGs from *Cx3cr1*GFP/+ or *Cx3cr1*GFP/GFP mice at 7 days after spared nerve injury (SNI) analyzed by flow cytometry (n=7). (**C**) Representative confocal images of L4 DRG from *Cx3cr1*GFP/+ or *Cx3cr1*GFP/GFP mice after 7 days of SNI. CX3CR1+ cells are shown

*Figure 7 continued on next page*

*Figure 7 continued*

in green and Iba1$^+$ cells are shown in magenta. Scale bars: 50 µm. (**D**) Quantification of CX3CR1$^+$ and Iba1$^+$ cells in DRGs at 7 days after SNI (n=5). (**E**) Mechanical allodynia was evaluated by paw withdrawal frequency using 0.008 g von Frey filament in *Cx3cr1$^{+/+}$* or *Cx3cr1$^{GFP/GFP}$* mice. Results are shown as the mean ± SEM. p-values were determined by one-way ANOVA followed by Bonferroni's post hoc test. *p<0.05; ns, not significant. Data are representative of at least two independent experiments.

The online version of this article includes the following figure supplement(s) for figure 7:

**Figure supplement 1.** Macrophage proliferation is impaired in the absence of CX3CR1 signaling.

have shown that CX3CL1/CX3CR1 signaling seems to be also involved in sensory ganglia macrophage activation/proliferation. In fact, in the DRG, the CX3CR1-proliferating macrophages are in close contact with the cell body of sensory neurons, which constitutively express the membrane-bound CX3CL1 (*Kim et al., 2011c*; *Huang et al., 2014*). Moreover, after peripheral nerve injury, membrane-bound CX3CL1 is reduced in the cell bodies of sensory neurons, suggesting their release and action in the sNAMs (*Zhuang et al., 2007*). Since the increase in the number of sNAMs was only partially reduced in CX3CR1 deficient mice, it is plausible that other signaling pathways are involved in the activation/proliferation of sNAMs in the DRGs after peripheral nerve injury. One possibility that has been recently explored is the CSF1-CSF1R signaling. In fact, injured neurons produce and release CSF-1 that in turn promotes sNAMs expansion in sensory ganglia through CSF1R activation (*Guan et al., 2016*; *Yu et al., 2020*).

Like classic cells in the innate immune system, sNAMs in the sensory ganglia also express Toll-like receptors (TLRs) and nucleotide-binding cytoplasmic oligomerization (Nod)-like receptors (NLRs). Previous studies indicate that activation of sNAMs in the sensory ganglia, after peripheral nerve injury, depends on downstream signaling generated by the activation of TLR2, TLR4, and TLR9 (*Shen et al., 2017*; *Kim et al., 2011b*; *Luo et al., 2019*). We also demonstrated that NOD2 deficiency prevented the increase in the number of sNAMs in the DRGs after SNI (*Guimarães et al., 2019*). Considering the involvement of PRRs in the activation/proliferation of DRG-resident macrophages, future studies will be necessary to clarify how these cells recognize or respond to a peripheral nerve injury, which is assumed to be a sterile condition.

Finally, we also address the importance of sNAMs for the production of pro-inflammatory cytokines, especially TNF and IL-1b, which have been described as upregulated in the sensory ganglia after peripheral nerve injury (*Ji et al., 2014*). Although there is a consensus that peripheral nerve injury triggers the up-regulation of these cytokines in the sensory ganglia, their cellular source is not completely characterized and there are discrepancies in the literature. We provided evidence by using different approaches that allow us to strongly indicate that, at least, TNF and IL-1b, are produced mainly by sNAMs. The production of these cytokines by sNAMs in the sensory ganglia has been recently implicated in the pathophysiology of neuropathic pain (*Guimarães et al., 2019*; *Yu et al., 2020*). Nevertheless, our data indicate that sNAMs are not the source of IL-6 after peripheral nerve injury. These data are consistent with previous reports indicating that sensory neurons start to produce IL-6 after different nerve injuries (*Murphy et al., 1995*; *Hu et al., 2020*). Furthermore, our data show that CX3CR1 signaling is not involved in the IL-6 production after peripheral nerve injury, further support the independence of its production by sNAMs.

In summary, the present study elucidates that the increase of sNAMs in the DRG triggered by peripheral nerve injury is a result of resident CX3CR1$^+$ macrophage proliferation and does not depend on blood-derived monocyte infiltration. This process seems to be important for the development of neuropathic pain caused by peripheral nerve injury. Furthermore, CX3CR1 signaling in sNAMs mediates their activation/proliferation, as well as the production of pro-inflammatory mediators. In conclusion, our findings might be useful to explore the modulation of sNAMs proliferation in conditions related to peripheral nerve injuries such as neuropathic pain.

# Methods

**Key resources table**

| Reagent type (species) or resource | Designation | Source or reference | Identifiers | Additional information |
|---|---|---|---|---|
| Antibody | Iba1 (Rabbit polyclonal) | Wako Chemicals | 019–19,741 | 1:400 |
| Antibody | Ki-67 (Rabbit monoclonal) | Abcam | ab16667 | 1:400 |
| Antibody | Anti-Rabbit IgG Alexa Fluor 594 (Goat-polyclonal) | Invitrogen | A11012 | 1:800 |
| Antibody | Anti-Rabbit IgG Alexa Fluor 647 (Chicken-polyclonal) | Invitrogen | A21443 | 1:800 |
| Antibody | CD45-BV421 (Rat-IgG2b monoclonal) | eBiosciences | 563890 | 1:350 |
| Antibody | CD11b-FITC (Rat-IgG2b monoclonal) | eBiosciences | 553310 | 1:250 |
| Antibody | CX3CR1-PE (Rat-IgG2a-K monoclonal) | Biolegend | 14906 | 1:250 |
| Antibody | Ly6G-APC (Rat-IgG2b monoclonal) | eBiosciences | 17-9668-82 | 1:250 |
| Antibody | Ly6C-PERCP (Rat-IgG2b monoclonal) | eBiosciences | 45-5932-82 | 1:250 |
| Chemical Compound | PBS, pH 7.4 (1x) | Sigma-Aldrich | 10010023 | |
| Chemical Compound | Xylazine | Akorn | NDC59399-110-20 | |
| Chemical Compound | Ketamine | Par Pharmaceutical | NDC42023-115-10 | |
| Chemical Compound | Isoflurane | Cristália | 667940172 | |
| Chemical Compound | TRIzol | Thermo Fischer Scientific | 15596026 | |
| Chemical Compound | High-Capacity cDNA | Thermo Fischer Scientific | 4368814 | |
| Chemical Compound | SYBR Green Master Mix | Thermo Fischer Scientific | | |
| Chemical Compound | RPMI 1640 Medium | Gibco | 11875093 | |
| Chemical Compound | Ara-C | Sigma-Aldrich | 69-74-9 | |
| Chemical Compound | aCSF | Tocris | 3525 | |
| Chemical Compound | Collagenase type 2 | Worthington Biochemical Corporation | 9001-12-1 | |
| Chemical Compound | Fixable Viability Dye eF780 | Thermo Fischer Scientific | 65-0865-14 | |
| Chemical Compound | Triton X-100 | Sigma-Aldrich | T8787 | |
| Chemical Compound | Bovine Seruim Albumin (BSA) | Sigma-Aldrich | A7906-100G | |
| Chemical Compound | Sucrose | Fisher Scientific | S5-500 | |
| Chemical Compound | Tissue Tek | Electron Microscopy Sciences | 62550–01 | |
| Chemical Compound | Veet Hair Remover | Reckitt Benckiser | | |
| Software, algorithm | StepOne Real-Time PCR System | Thermo Fischer Scientific | 4376357 | |

*Continued on next page*

*Continued*

| Reagent type (species) or resource | Designation | Source or reference | Identifiers | Additional information |
|---|---|---|---|---|
| Software, algorithm | FACSAria III Cell Sorter | BD Biosciences | | |
| Software, algorithm | FACSVerse | BD Biosciences | | |
| Software | Prism | GraphPad | Version 8 | |
| Software | FlowJo | FlowJo | V10.8.1 | |
| Genetic Reagent (*Mus musculus* C57BL/6) | *Cx3cr1*<sup>GFP</sup> | Jackson Laboratories | Strain# 005582 | PMID:17944871 |
| Genetic Reagent (*Mus musculus* C57BL/6) | *Ccr2*<sup>RFP</sup> | Jackson Laboratories | Strain# 017586 | PMID:21060874 |
| Genetic Reagent (*Mus musculus* C57BL/6) | eGFP C57BL/6-(Tg[CAG-EGFP]) | Jackson Laboratories | Strain# 006567 | PMID:9175875 |
| Genetic Reagent (*Mus musculus* C57BL/6) | *Ms4a3*<sup>Cre</sup> | *Liu et al., 2019*. | N/A | PMID:31491389 |
| Genetic Reagent (*Mus musculus* C57BL/6) | *Rosa26tdTomato* | Jackson Laboratories | Strain# 007914 | PMID:20023653 |
| Sequence-based reagent | Gapdh_F | Sigma-Aldrich (This paper) | PCR primers | GGGTGTGAACCACGAGAAAT |
| Sequence-based reagent | Gapdh_R | Sigma-Aldrich (This paper) | PCR primers | CCTTCCACAATGCCAAAGTT |
| Sequence-based reagent | Aif1_F | Sigma-Aldrich (This paper) | PCR primers | GCTTCAAGTTTGGACGGCAG |
| Sequence-based reagent | Aif1_R | Sigma-Aldrich (This paper) | PCR primers | TGAGGAGCCATGAGCCAAAG |
| Sequence-based reagent | Cx3cr1_F | Sigma-Aldrich (This paper) | PCR primers | GCCTCTGGTGGAGTCTGCGTG |
| Sequence-based reagent | Cx3cr1_R | Sigma-Aldrich (This paper) | PCR primers | CGCCCAAATAACAGGCCTCAGCA |
| Sequence-based reagent | Cx3cl1_F | Sigma-Aldrich (This paper) | PCR primers | CGCGTTCTTCCATTTGTGTA |
| Sequence-based reagent | Cx3cl1_R | Sigma-Aldrich (This paper) | PCR primers | CTGTGTCGTCTCCAGGACAA |
| Sequence-based reagent | Il6_F | Sigma-Aldrich (This paper) | PCR primers | TTCCTACCCCAATTTCCAAT |
| Sequence-based reagent | Il6_R | Sigma-Aldrich (This paper) | PCR primers | CCTTCTGTGACTCCAGCTTATC |
| Sequence-based reagent | Tnf_F | Sigma-Aldrich (This paper) | PCR primers | GCCACAAGCAGGAATGAGAAG |
| Sequence-based reagent | Tnf_R | Sigma-Aldrich (This paper) | PCR primers | AGCAAGCAGCCAACCAGG |
| Sequence-based reagent | Kcnj1_F | Sigma-Aldrich (This paper) | PCR primers | GGGCTATCAGAGGCTGTGTC |
| Sequence-based reagent | Kcnj1_R | Sigma-Aldrich (This paper) | PCR primers | GTGACAGGCAAACTGCTTCA |
| Sequence-based reagent | Il1b_F | Sigma-Aldrich (This paper) | PCR primers | TGACAGTGATGATGAGAATGACCTGTTC |

*Continued on next page*

*Continued*

| Reagent type (species) or resource | Designation | Source or reference | Identifiers | Additional information |
|---|---|---|---|---|
| Sequence-based reagent | Il1b_R | Sigma-Aldrich (This paper) | PCR primers | TTGGAAGCAGCCCTTCATCT |
| Sequence-based reagent | Cfs1r_F | Sigma-Aldrich (This paper) | PCR primers | ACACGCACGGCCACCATGAA |
| Sequence-based reagent | Cfs1r_R | Sigma-Aldrich (This paper) | PCR primers | GCATGGACCGTGAGGATGAGGC |

## Animals

For all experiments, we use 7–10 week-old males, unless specified in the text. C57BL/6 wild-type (WT) mice were purchased from Jackson Laboratory, bred, and raised in-house. *Ccr2*$^{RFP/RFP}$ mice (*Saederup et al., 2010*), and *Cx3cr1*$^{GFP/GFP}$ mice (*Jung et al., 2008*). *Ccr2*$^{RFP/+}$-*Cx3cr1*$^{GFP/+}$ mice were generated by crossbreeding *Ccr2*$^{RFP/RFP}$ mice with *Cx3cr1*$^{GFP/GFP}$ mice. We also used hemizygous transgenic mice expressing eGFP, C57BL/6-(Tg[CAG-EGFP]), under the control of the chicken-actin promoter and cytomegalovirus enhancer (*Okabe et al., 1997*). M*s4a3*$^{Cre/−}$ mice have been previously described (*Liu et al., 2019*) and kindly provided by Dr. Florent Ginhoux, (Singapore Immunology Network, Agency for Science, Technology and Research, Singapore). Rosa26$^{tdTomato}$ reporter mice have been previously described (*Madisen et al., 2010*).

Local colonies of transgenic mice were then established and maintained on a C57BL/6 background at the animal care facility of the Ribeirao Preto Medical School, University of Sao Paulo. Food and water were available ad libitum. Animal care and handling procedures were under the guidelines of the International Association for the Study of Pain for those animals used in pain research and were approved by the Committee for Ethics in Animal Research of the Ribeirao Preto Medical School—University of São Paulo (Process number 002/2017).

## SNI model

SNI was used as a model of a distal peripheral nerve injury. Briefly, animals were anesthetized with 1% isoflurane (v/v), and the sciatic nerve and its 3 terminal branches were exposed. The tibial and common peroneal branches were ligated using 5–0 silk and sectioned distally, whereas the sural nerve remained intact, as previously described (*Decosterd and Woolf, 2000*). Finally, the skin was sutured in two stitches.

## HSV-1 infection

Mice were anesthetized with 1% isoflurane (v/v) and then the mid flank and right foot were clipped and depilated with a chemical depilatory (Veet Hair Remover; Reckitt Benckiser). Three days later, HSV-1 (2 × 10$^5$ PFUs in 20 ul) was inoculated on the skin of the right hind paw (5 × 5 mm), after the skin was scarificated with sandpaper. The virus was applied directly to the scarified area (*Silva et al., 2017*).

## Mechanical pain hypersensitivity measurements (mechanical allodynia)

Animals were acclimated to the behavioral platform for 60 min before measuring nociceptive responses. The mechanical nociceptive threshold was evaluated in *Cx3cr1*$^{+/+}$, *Cx3cr1*$^{GFP/+}$, *Cx3cr1*$^{GFP/GFP}$ mice (n=6), that were placed on an elevated wire grid, and the plantar surface of the ipsilateral and contralateral hind paws were stimulated perpendicularly with of von Frey filaments (Stoelting, Chicago, IL, USA). Withdrawal frequency assay was performed by stimulating the ipsilateral and contralateral hind paws 10 times with a 0.008 g von Frey filament. Data are presented as the percentage of paw withdrawal.

## Drugs administration

To inhibit the proliferation of sNAMs in the lumbar DRGs, animals were treated with Ara-C (Cytarabine, Sigma, St. Louis, MO, USA). On days 2, 3, 4, and 5 after SNI, the animals received aCSF (Tocris, USA) or Ara-C (50 ug/µl) intrathecally (*Wang et al., 2005*). Under 1% isoflurane (v/v) anesthesia, mice were securely held in one hand by the pelvic girdle and inserted a BD Ultra-Fine (29 G) insulin syringe (BD, Franklin 6 Lakes, NJ, USA) directly on subarachnoid space (close to L4–L5 segments) of the spinal

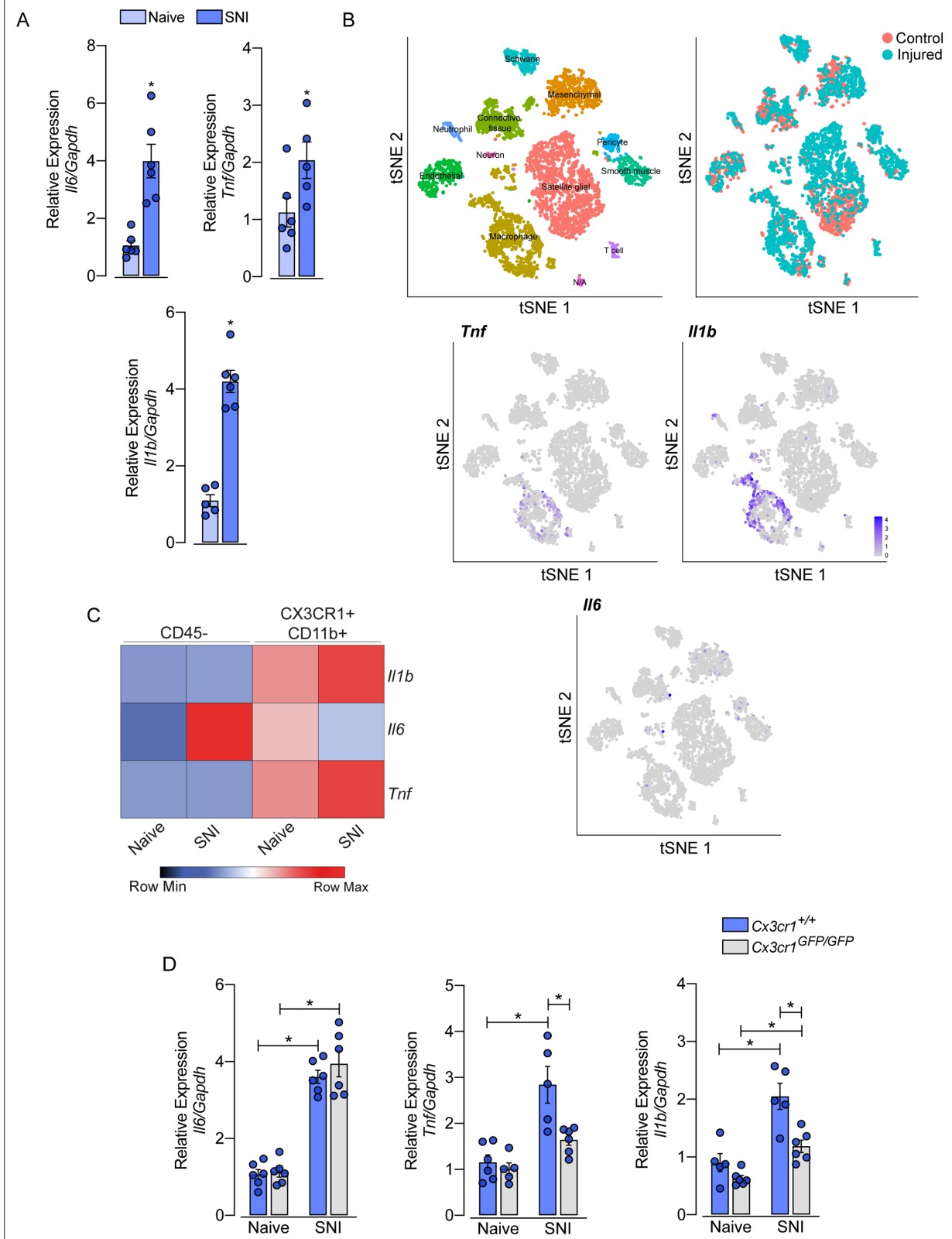

**Figure 8.** Role of sensory neuron-associated macrophages (sNAMs) in the production of pro-inflammatory cytokines in the sensory ganglia after spared nerve injury (SNI). (**A**) RT-qPCR analysis of *Il6, Il1b,* and *Tnf* mRNA expression relative to *Gapdh* in the dorsal root ganglia (DRG) from WT mice naive or 7 days after SNI (n=5–6). (**B**) t-SNE plot analysis showing clusters of cell populations (GSE139103) and expression profile of *Tnf, Il1b, and Il6* in the DRGs after peripheral nerve injured or naive mice. (**C**) RT-qPCR analysis of *Il6, Il1b,* and *Tnf* mRNA expression relative to *Gapdh* in CD45⁻ or CX3CR1⁺CD11b⁺

*Figure 8 continued on next page*

Figure 8 continued

cells isolated from *Cx3cr1*[GFP/+] mice DRGs at 7 days after SNI or naive (n=8 pooled). (**D**) RT-qPCR analysis of *Il6*, *Il1b*, and *Tnf* mRNA expression relative to *Gapdh* in the DRGs from *Cx3cr1*[+/+] or *Cx3cr1*[GFP/GFP] mice after 7 days after SNI (n=5–6). Results are shown as the mean ± SEM. p-values were determined by (**A**) two-tailed Student's *t*-test and (**D**) one-way ANOVA followed by Bonferroni's post hoc test. *p<0.05. Data are representative of at least two independent experiments.

The online version of this article includes the following figure supplement(s) for figure 8:

**Figure supplement 1.** Representative gating strategies for flow cytometry analysis and cell sorting.

cord. A sudden lateral movement of the tail indicated the proper placement of the needle in the intrathecal space. Then, the syringe was held in a specific position for a few seconds and progressively removed to avoid any outflow of the substances. For all administrations, 5 µl of volume was used. Mechanical allodynia was evaluated 4 hr after the injections.

## Quantitative real-time RT-PCR

At the indicated time points after peripheral nerve injury (SNI) or naive mice were terminally anesthetized with xylazine/ketamine and then transcardially perfused with phosphate-buffered solution (PBS 1 x). The dorsal root ganglia (L3-L5) ipsilateral to the lesion were collected and rapidly homogenized in 500 µl of TRIzol solution (Thermo Fischer Scientific) reagent at 4°C. Then, total cellular RNA was purified from the tissue according to the manufacturer's instructions. The purified total RNA was measured by a spectrophotometer using the wavelength absorption ratio (260/280 nm), which was between 1.8 and 2.0 for all preparations. The obtained RNA samples were reverse-transcribed with High Capacity Kit (Thermo Fischer Scientific). Real-time PCR was performed using specific primers for the mouse genes *Aif1*, *Cx3cr1*, *Il1b*, *Tnf*, *Il6*, *Cx3cl1*, *Csf1r, and Kcnj1* (Key resources table). The levels of each gene were normalized to the expression of *Gapdh*. Reactions were conducted on the Step One Real-Time PCR System using the SYBR-green fluorescence system (Applied Biosystems, Thermo Fisher Scientific, Waltham, MA, USA). The results were analyzed by the quantitative relative expression $2^{-\Delta\Delta Ct}$ method as previously described (*Livak and Schmittgen, 2001*).

## Immunofluorescence analysis and quantification

At the indicated times after nerve injury, mice were terminally anesthetized with xylazine/ketamine and transcardially perfused with PBS 1x. Following by 4% paraformaldehyde (PFA) in 0.1 M PBS, pH 7.4 (4oC). After the perfusion, the dorsal root ganglia (L3-L5) were dissected, post-fixed in PFA for 2 h, and then bathed in 30% sucrose overnight. The DRGs were covered with tissue Tek O. C. T. (Electron Microscopy Sciences, 62550–01) and sections were cut (16 $\mu$m) in a cryostat (Leica Biosystems, CM3050S). Then, the sections were incubated in a blocking buffer, 2% BSA and 0.3%T riton X-100 in PBS. After 1 h, the sections were incubated overnight at 4oC with polyclonal primary antibodies for ionized calcium-binding adapter molecule 1 (Iba1; 1:400; Wako Chemicals, Richmond, VA, USA) or antibody for proliferation marker (Ki-67; 1:400; Abcam, Cambridge, UK). The sections were washed with PBS and incubated with the appropriate secondary antibody solution for 2h at room temperature (IgG conjugated Alexa Fluor 594 and/or 647; 1:800; Invitrogen, Carlsbad, CA). The sections were washed with PBS and mounted with coverslips adding Aqueous Mounting Medium, Fluoroshield with DAPI (Sigma-Aldrich). For the evaluation of CX3CR1 and CCR2 expression, the aforementioned genetically modified Ccr2RFP/+: Cx3cr1GFP/+ mice were used.

The DRG sections were acquired on Zeiss LSM780 confocal microscope. Images were processed in the FIJI package for ImageJ software and the quantification of macrophages was performed using the cell counter. For all measurements, three sections of each DRG (L4 or L5) were analyzed, and the results were averaged to generate the value for a single mouse. Cells quantifications were normalized by the areas (mm²) of each DRG tissue of individual slices.

## Flow cytometry and cell sorting

DRGs (L3-L5) were collected from naive or SNI mice and incubated in a solution of 1 ml of RPMI 1640 medium with 2 mg/ml of collagenase type II (Worthington Biochemical Corporation) for 30 min at 37°C. After digestion, the DRGs were mechanically ground through 40 µm cell strainer, and the cell suspension was washed with PBS 1 x. The cells were resuspended in PBS 1 x containing specific

monoclonal antibodies against surface markers for 10 min at room temperature. Dead cells were excluded by Fixable Viability Dye (Catalog number 65-0865-14, Thermo Fisher Scientific, 1:3000). The following monoclonal antibodies were used: anti-CD45-BV421 (Clone 30-F11, BD Biosciences, 1:350), anti-CD11b-FITC (Clone M1/70, BD Biosciences, 1:250), anti-Ly6G-APC (Clone 1A8, BD Biosciences, 1:250), anti-Ly6C-PERCP (Clone HK1.4, eBiosciences, 1:20) and anti-CX3CR1-PE (Clone SA011F11, BioLegend, 1:250). The sample acquisition was performed by FACSVerse flow cytometry instrument (BD Biosciences, San Jose, CA, USA), and data were analyzed using FlowJo software BD (Becton, Dickinson & Company, USA).

## Cell sorting

DRGs were collected from naive or SNI *Cx3cr1*$^{GFP/+}$ mice (DRGs from eight mice were pooled). As described above, the tissues were digested and filtered through a cell strainer. Samples were then centrifuged, and the supernatant was discarded. Cellular pellet was resuspended in a solution containing cell surface markers (CD11b, CD45, and Live/Dead), stained for 10 min at room temperature, and then further sorted as macrophages (CX3CR1$^+$CD11b$^+$) in a FACS Aria III sorter. The full-gating strategy used to perform cell sorting and all flow cytometry experiments from sensory ganglia is depicted in *Figure 8—figure supplement 1*. Sorted cells were submitted to RNA extraction (using the RNeasy Micro Kit - Qiagen), reverse-transcribed with High Capacity Kit (Thermo Fischer Scientific), and analyzed by RT-PCR with a Step One Real-time PCR system as described above (Applied Biosystems).

## Parabiosis

Parabiosis was performed as previously described (*Kamran et al., 2013*). Briefly, 8-week-old matched female WT and C57BL/6-(Tg[CAG-EGFP]) mice were co-housed for 2 weeks to reduce stress and then surgically attached for 1 month. Then, they were deeply anesthetized with 1% isoflurane (v/v), and a skin incision was made along the contiguous flanks on the prepared side of each animal. Two animals were paired through the skin, each mouse was sutured to each other, enabling a shared circulation between the two mice. 30 days after the recovery from the parabiosis surgery, mice were subjected to the SNI model and surgically separated 7 days after. Blood exchange was confirmed upon separation by examining GFP$^+$ cells in the bloodstream of WT mice by flow cytometry.

## Re-analysis of public scRNA-seq data

The scRNA-seq data from mice DRGs was acquired from the Gene Expression Omnibus (GEO) database under the series number GSE139103 (*Avraham et al., 2020*). The single-cell libraries were generated using GemCode Single-Cell 3′ Gel Bead and Library Kit on the 10 X Chromium system (10 X Genomics). The dataset contains cells from four animals, of which two are naive mice and two with injured DRG. The feature barcode matrix was analyzed using Seurat v3. The cells were filtered according to the criteria: 600–10000 total reads per cell, 500–4000 expressed genes per cell, and mitochondrial reads <10%. Clusters were identified using shared nearest neighbor (SNN) based clustering based on the first 30 PCAs and resolution = 0.5. The same principal components were used to generate the t-SNE projections. Differentially expressed genes between samples from naive and injured mice for each cluster were identified using FDR <0.05 and |avg_log2FC|>0.25.

## Experimental study design (statistics details)

The n sample was determined based on previous publications and/or internal pilot data, to be adequate for statistical analysis and ensured reproducibility. No statistical methods were used to determine the sample size. Additionally, experimental groups were blinded during qualifications. Data are reported as the mean ± SEM. Result analysis was performed by One-way ANOVA followed by the Bonferroni test (for three or more groups) comparing all pairs of columns. Alternatively, an unpaired Student's *t*-test was used to compare two different groups. Values of $p<0.05$ were considered statically significant. Statistical analysis was performed with GraphPad Prism 8 software.

## Acknowledgements

The authors gratefully acknowledge the technical assistance of Ieda Santos, Roberta Rosales, Denise Brufato, Marco Antônio Ribeiro, and Katia Santos. We would like to thank Simone Brioschi (Washington

University in St. Louis) for providing good suggestions for the paper and also the *Ccr2* knockout mice. We thank Florent Ginhoux (Singapore Immunology Network, Agency for Science, Technology and Research) for kindly provided the *Ms4a3*[Cre] mice. The research leading to these results has received funding from the São Paulo Research Foundation (FAPESP) under grant agreement n ∘ 2013/08216–2 (Center for Research in Inflammatory Disease); from Coordenação de Aperfeiçoamento de Pessoal de Nível Superior (CAPES).

## Additional information

### Funding
No external funding was received for this work.

### Author contributions

Rafaela M Guimarães, Formal analysis, Validation, Investigation, Visualization, Writing – original draft; Conceição E Aníbal-Silva, Conceptualization, Formal analysis, Investigation, Methodology, Writing – review and editing; Marcela Davoli-Ferreira, Data curation, Formal analysis, Investigation, Methodology; Francisco Isaac F Gomes, Miriam M Fonseca, Larissa P Andrade, Formal analysis, Investigation, Visualization; Atlante Mendes, Data curation, Formal analysis, Investigation, Visualization; Maria CM Cavallini, Methodology; Samara Damasceno, Formal analysis, Visualization; Marco Colonna, Cyril Rivat, Resources, Writing – review and editing; Fernando Q Cunha, Resources, Funding acquisition, Investigation, Visualization; José C Alves-Filho, Funding acquisition, Visualization, Writing – review and editing; Thiago M Cunha, Conceptualization, Resources, Data curation, Formal analysis, Supervision, Funding acquisition, Validation, Investigation, Visualization, Methodology, Writing – original draft, Project administration, Writing – review and editing

### Author ORCIDs

Rafaela M Guimarães http://orcid.org/0000-0002-2182-4215
Conceição E Aníbal-Silva http://orcid.org/0000-0001-5562-0205
Thiago M Cunha http://orcid.org/0000-0003-1084-0065

### Ethics
Animal care and handling procedures were under the guidelines of the International Association for the Study of Pain for those animals used in pain research and were approved by the Committee for Ethics in Animal Research of the Ribeirao Preto Medical School- University of São Paulo (Process number 002/2017).

### Decision letter and Author response
Decision letter https://doi.org/10.7554/eLife.78515.sa1
Author response https://doi.org/10.7554/eLife.78515.sa2

## Additional files

### Supplementary files
• MDAR checklist

### Data availability
All data generated or analyzed during this study are included in the manuscript. Public scRNA-seq data are available in Gene Expression Omnibus (GEO) database under the series number GSE139103 (*Avraham et al., 2020*).

The following previously published dataset was used:

| Author(s) | Year | Dataset title | Dataset URL | Database and Identifier |
|---|---|---|---|---|
| Avraham O, Deng PY, Jones S, Kuruvilla R, Semenkovich CF, Klyachko VA, Cavalli V | 2020 | Satellite glial cells promote regenerative growth in sensory neurons | https://www.ncbi.nlm.nih.gov/geo/query/acc.cgi?acc=GSE139103 | NCBI Gene Expression Omnibus, GSE139103 |

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
