## [Editor Report]

Guimaraes et al. address the origin of the macrophage increase in sensory ganglia after peripheral nerve injury. The authors show that there is no major influx by blood-derived monocytes into ganglia after injury and that resident macrophages proliferate, which is dependent on CX3CR1 signaling. Overall the work is clear and sound and should be of interest to immunologists and neurobiologists.

---

## [Decision Letter]

**Decision letter after peer review:**

Thank you for submitting your article "Sensory neuron-associated macrophages proliferate in the sensory ganglia after peripheral nerve injury in a CX3CR1 signaling dependent manner" for consideration by *eLife*. Your article has been reviewed by 3 peer reviewers, and the evaluation has been overseen by a Reviewing Editor and Satyajit Rath as the Senior Editor. The following individuals involved in the review of your submission have agreed to reveal their identity: Richard E. Zigmond (Reviewer #2); Rebecca Gentek (Reviewer #3).

Essential revisions:

1) Reviewers expressed concerns about novelty as similar or identical experiments have already been published (Kalinski et al. 2020 using parabiosis and Iwai et al. 2021 in J Neuroinflammation using irradiation with a metal head guard followed by transplantation of hematopoietic cells. Yu et al. (in Nature Communications) using Ki67 and Iwai et al. 2021 (cited above) using BrDU showed that there is a proliferation of resident macrophages in sensory ganglia after a nerve lesion).

However, a novel insight is that proliferation of macrophages is not seen in CX3CR1 knockout animals. The authors should develop this aspect and its significance, as well as perform more than a single experiment about the role of CX3CR1 in macrophage proliferation.

The authors should investigate the involvement of the ligand CX3CL1 and evaluate if proliferation was normal in another knockout animal, i.e., CCR2 -/-. Finally, the most important question that needs to be asked is whether proliferation of resident macrophages is required for some functional aspects of nerve regeneration.

2) Provide important controls and missing experiments as recommended by the reviewers.

*Reviewer #1 (Recommendations for the authors):*

– The authors should correlate CX3CR1 and Iba1 expression for their definition of resident macrophages.

– The authors show that there is no increase in CCR2 expression in DRG at 3dpi, however, monocytes could downregulate CCR2 very rapidly. Additionally, they also express CX3CR1-GFP. Please add 1dpi in Figure 2B and C.

– What is the reason that the microscopy data (Figure 2C) and the flow cytometry data (Figure S2B) correlate so poorly? In microscopy, the CX3CR1+ cells are shown to triple at 7dpi while in flow cytometry there is only a minor increase at the same time point. The authors should verify their gating strategy and extraction protocol.

– The data from CCR2ko mice are important and should be moved to the main figures. It would also be good to show microscopy data or flow cytometry data from these mice as well to show that indeed macrophages are similar and not just CX3CR1 expression.

– For figure 6, DRGs from CX3CR1ko mice should also be imaged and quantified to show the absent increase of macrophages post-injury, especially since data from these two methods differed above.

– In Figure 7, CD11b+CX3CR+ macrophages were compared to CD45neg cells. What about all other CD45+ and CD11b+CX3CR1- cells? One would expect that those could also produce proinflammatory cytokines. In order to be able to conclude that it is only the resident macrophages, the authors have to sort the other cell populations and compare their expression as well. The authors should also show if there is any infiltration by granulocytes after injury, with regard to their discussion of the results.

*Reviewer #2 (Recommendations for the authors):*

In Figures5, 6, and 7, the meaning of the number sign (#) is not defined. It would also be helpful if the comparisons being indicated by an asterisk would be defined when there are more than two groups.

In most of the figure legends, it is stated that "Error bars show the mean +/- S.E.M." Don't the error bars actually just show the S.E.M.?

A few of the references seem out of place. Does Wang et al. 2020 show repair or neuropathic pain?

*Reviewer #3 (Recommendations for the authors):*

I would like to congratulate the authors on their work, which is overall clear and sound.

---

## [Author Response]

Essential revisions:1) Reviewers expressed concerns about novelty as similar or identical experiments have already been published (Kalinski et al. 2020 using parabiosis and Iwai et al. 2021 in J Neuroinflammation using irradiation with a metal head guard followed by transplantation of hematopoietic cells. Yu et al. (in Nature Communications) using Ki67 and Iwai et al. 2021 (cited above) using BrDU showed that there is a proliferation of resident macrophages in sensory ganglia after a nerve lesion).However, a novel insight is that proliferation of macrophages is not seen in CX3CR1 knockout animals. The authors should develop this aspect and its significance, as well as perform more than a single experiment about the role of CX3CR1 in macrophage proliferation.

First of all, we would like to thank the editor and referees for the opportunity to review our manuscript. Certainly, all comments were very important to improve the new version of our manuscript. As suggested by the editor and referees, we have performed a substantial number of additional experiments to further confirm our hypothesis that the increase of macrophages in the DRGs after peripheral nerve injury (SNI) is not due to the infiltration of blood monocytes, but it is caused by the expansion/proliferation of resident CX3CR1^+^ macrophages. Among the additional experiments, we are providing in the new version of the manuscript: (1) A series of additional experiments showing that CCR2 is not important for the increase of macrophages in the DRGs after peripheral nerve injury (new Figure 3); (2) Additional experiment using Ms4a3^Cre-tdTomato^ animals confirmed that blood monocytes (and neutrophils) did not infiltrate significantly the DRGs after SNI (new Figure 5); (3) Additional experiment showing no change in Ly6C^+^ monocytes in the DRGs after nerve injury (new Figure 2—figure supplement 3) and others. Related to the role of CX3CR1 signalling on the proliferation of DRGs’ resident macrophages after peripheral nerve injury, we are also providing additional data, such as the analyses of: (1) Iba1 expression by IF (new Figure 7); (2) Ki67 expression by IF in CX3CR1 KO mice (new Figure 2—figure supplement 6).

The authors should investigate the involvement of the ligand CX3CL1 and evaluate if proliferation was normal in another knockout animal, i.e., CCR2 -/-. Finally, the most important question that needs to be asked is whether proliferation of resident macrophages is required for some functional aspects of nerve regeneration.

Regarding these points, we have evaluated the expression of CX3CL1 in the DRGs after peripheral nerve injury, which was up-regulated (new figure 7A). We also evaluated the number of macrophages in the DRGs of CCR2 KO mice after SNI. Corroborating the data, we have previously shown by PCR (old figure S3, new figure 3A), the expansion of CX3CR1^+^ macrophages in DRGs after SNI was similar in CCR2 KO compared with WT (please see new figure 3). Finally, we have evaluated the function of macrophage proliferation in the DRGs for the development of neuropathic pain. We have found that the AraC (cell proliferation inhibitor) reduced SNI-induced mechanical allodynia, which was associated with a prevention of macrophage increase in the DRGs after SNI (please see new figure 6).

2) Provide important controls and missing experiments as recommended by the reviewers.

We have attended to all the referee’s suggestions.

Reviewer #1 (Recommendations for the authors):– The authors should correlate CX3CR1 and Iba1 expression for their definition of resident macrophages.

This was a great suggestion. By using Cx3cr1^GFP/+^ mice, we are now providing images of DRGs after 7 days of sciatic nerve injury stained with Iba1. Our results reveal that ~ 97 % CX3CR1^+^ cells are Iba1+ and vice-versa. Please see the new Figure 2—figure supplement 1A.

– The authors show that there is no increase in CCR2 expression in DRG at 3dpi, however, monocytes could downregulate CCR2 very rapidly. Additionally, they also express CX3CR1-GFP. Please add 1dpi in Figure 2B and C.

We agree with the referee and the suggested experiment was performed. Our results showed that there was no increase in the number of CX3CR1^+^ and CCR2^+^ cells 1 day after SNI. These results are presented in the new figure 2A-B.

– What is the reason that the microscopy data (Figure 2C) and the flow cytometry data (Figure S2B) correlate so poorly? In microscopy, the CX3CR1+ cells are shown to triple at 7dpi while in flow cytometry there is only a minor increase at the same time point. The authors should verify their gating strategy and extraction protocol.

Thank you for the comment. Our protocol has been improved in the course of this study. From a certain point, our protocol for tissue harvesting, digestion and staining was optimised, and the following experiments were better. Nevertheless, we repeated this mentioned experiment, as you can see in the new figure 2D.

– The data from CCR2ko mice are important and should be moved to the main figures. It would also be good to show microscopy data or flow cytometry data from these mice as well to show that indeed macrophages are similar and not just CX3CR1 expression.

Thanks for the excellent suggestions. In the new version of the manuscript, we performed several additional experiments, including immunofluorescence and flow cytometry analyses of macrophages population in DRGs from CCR2^-/-^ mice, 7 days after SNI. We found that the increase in macrophage numbers does not change in the absence of CCR2 receptors, which corroborates our previous data. We also observed that, at 3 days after SNI, the increase in Ki67^+^macrophages in the DRGs from CCR2^-/-^ mice is also similar to DRGs from WT mice. These new data are now in figures 3 and Figure 6—figure supplement 5.

– For figure 6, DRGs from CX3CR1ko mice should also be imaged and quantified to show the absent increase of macrophages post-injury, especially since data from these two methods differed above.

We agree with the referee and the suggested experiment was performed. Corroborating our flow cytometry analyses, immunofluorescence analyses of CX3CR1^+^/Iba1^+^ macrophages in the DRGs from Cx3Cr1^GFP/GFP^ mice, 7 days after injury revealed a reduction compared to DRGs from Cx3Cr1^GFP/+^. We also analysed Ki67^+^macrophages in the DRGs from Cx3cr1^GFP/GFP^ mice, 3 days after SNI, which was also reduced compared to what we have obtained in the DRGs from Cx3cr1^GFP/+^ mice. Please see new figures 7 and Figure 7—figure supplement 6.

– In Figure 7, CD11b+CX3CR+ macrophages were compared to CD45neg cells. What about all other CD45+ and CD11b+CX3CR1- cells? One would expect that those could also produce proinflammatory cytokines. In order to be able to conclude that it is only the resident macrophages, the authors have to sort the other cell populations and compare their expression as well. The authors should also show if there is any infiltration by granulocytes after injury, with regard to their discussion of the results.

Thank you for the comment. In fact, CX3CR1+ macrophages correspond to more than 90% of leukocytes in the DRGs (in homeostasis and also after injury). So even if there is some other population of CD45^+^ cells in this tissue, this would be negligible compared to the macrophage population. Even these remaining cells (some are neutrophils) are possible contamination with leptomeninges that are attached to the harvested DRGs. Several times we tried to stain neutrophils in the parenchyma of the DRG and we never observed any staining with different markers. On the other hand, when we perform flow cytometry, we can observe that there is a population of CD11b^+^Ly6G^+^ cells (see new figure 1E), although this neutrophil population did not increase after injury and may be a remaining cells present in the meninges that surround the DRGs. Furthermore, the lack of increase in Ms4a3-tdTomato^+^ cells in the DRGs after SNI (see new figure 5), further supports the hypothesis that neutrophils did not infiltrate the DRGs after SNI.

Reviewer #2 (Recommendations for the authors):In Figures5, 6, and 7, the meaning of the number sign (#) is not defined. It would also be helpful if the comparisons being indicated by an asterisk would be defined when there are more than two groups.

We agree with the referee. These suggestions were incorporated into the new version of the manuscript. Please see new figures.

In most of the figure legends, it is stated that "Error bars show the mean +/- S.E.M." Don't the error bars actually just show the S.E.M.?

We corrected the legends in the new version of the manuscript.

A few of the references seem out of place. Does Wang et al. 2020 show repair or neuropathic pain?

We have checked the citations through the new version of the manuscript.